# Psychopathological Symptomatology and Sleep Quality in Chronic Primary Musculoskeletal Pain: A Comparison with Healthy Controls

**DOI:** 10.3390/healthcare13161965

**Published:** 2025-08-11

**Authors:** Alejandro Arévalo-Martínez, Carlos Barbosa-Torres, María Elena García-Baamonde, César Luis Díaz-Muñoz, Juan Manuel Moreno-Manso

**Affiliations:** 1Department of Psychology, Faculty of Education and Psychology, University of Extremadura, 06071 Badajoz, Spain; aarevaloj@unex.es (A.A.-M.); mgarsan@unex.es (M.E.G.-B.); jmmanso@unex.es (J.M.M.-M.); 2Department of Medical-Surgical Therapy, Faculty of Medicine and Health Sciences, University of Extremadura, 06071 Badajoz, Spain; cdiazmun@unex.es

**Keywords:** psychopathology, chronic pain, chronic primary musculoskeletal pain, sleep quality, pain intensity

## Abstract

Background/Objectives: Chronic musculoskeletal pain without a clearly identifiable medical cause is characterised by significant emotional distress and/or functional disability. Given the relatively limited research specifically addressing chronic primary musculoskeletal pain (CPMP), as defined in the latest revision of the International Classification of Diseases (ICD-11), the present study aimed to examine its psychopathological and sleep-related implications, and to explore whether pain intensity is associated with psychological distress and poor sleep quality. Methods: This observational study included 60 adult participants, comprising 30 patients diagnosed with CPMP and 30 healthy controls without any diagnosis. Participants completed the Numeric Pain Rating Scale (NPRS), the Symptom Checklist-90-R (SCL-90-R), and the Pittsburgh Sleep Quality Index (PSQI). Results: Patients with CPMP exhibited significantly higher levels of psychopathological symptomatology on the SCL-90-R and poorer sleep quality on the PSQI compared to controls (*p* < 0.05 for most dimensions). Greater pain intensity on the NPRS was strongly associated with psychological distress (e.g., GSI: r = 0.838, *p* < 0.01) and poor sleep quality (r = 0.785, *p* < 0.01). Hierarchical regression analyses revealed that pain intensity may play a meaningful role in both psychological distress and sleep quality (*p* < 0.05 across all models), even after statistically controlling for sex, age, and pain duration. Conclusions: These findings suggest that pain intensity is not only a key physical symptom, but also a relevant factor in understanding the broader psychological vulnerability in patients with CPMP. The present study contributes to a deeper understanding of the psychopathological and functional impact of CPMP and underscores the need for tailored psychological interventions to address the comorbid symptoms associated with this condition.

## 1. Introduction

Chronic musculoskeletal pain (CMP) is one of the most frequent causes of disability in the general population [1]. Globally, it is estimated that between 20% and 30% of individuals experience CMP [2,3]. This condition is characterised by an unpleasant sensory and emotional experience that persists or recurs for more than three months, originating in muscles, bones, joints, or tendons [4,5]. The most affected areas include the lower back, neck, and extremities. It can manifest in one or more of these regions and may occur in the absence of any identifiable underlying condition [4,6].

Among the various factors influencing chronic pain, pain intensity remains one of the most frequently studied [7,8]. Although the sensation of pain in CMP varies between individuals, it tends to be intense and localised in specific areas [1]. Importantly, pain intensity is not always proportional to the severity of the physical injury, suggesting that other variables—such as psychological symptoms—may influence the pain experienced [1,9,10]. Persistent pain can have significant psychosocial consequences, profoundly affecting the quality of life of those who suffer from it. Therefore, understanding the potential impact of comorbid symptoms in CMP is essential [1,11].

An increasing number of studies have adopted the biopsychosocial model to explain the symptoms and disabilities associated with CMP [6,12]. Within this framework, psychological factors are particularly relevant in the experience of pain, shifting the focus away from purely biomedical explanations [10,12]. Recent systematic reviews have reported elevated rates of psychopathological symptoms among individuals with CMP, including depression, anxiety, distress, somatisation, and sleep disturbances. These symptoms, when persistent over time, may increase the risk of pain chronification [11,13,14]. Among them, sleep disturbances are especially prevalent and disabling, and are increasingly recognised as a core component of the psychological burden associated with chronic pain [15,16]. Far from being a mere consequence of pain, recent studies suggest that poor sleep quality may intensify psychological distress by impairing emotional regulation and cognitive functioning, thereby reinforcing the cycle of pain and emotional suffering [17,18]. Nevertheless, the relationship between pain intensity and psychological symptomatology in CMP remains unclear. While some studies have found that psychological symptoms predict pain intensity, less is known about whether pain intensity itself can explain psychopathological symptoms [7,8,11,12].

The International Classification of Diseases (ICD-11) has aimed to clarify the conceptualisation of CMP, addressing limitations in earlier versions [5]. It introduces two diagnostic categories based on pain aetiology. Chronic primary musculoskeletal pain (CPMP) is diagnosed when pain persists for more than three months without a clearly identifiable medical cause and is accompanied by significant emotional distress and/or functional impairment [5,19]. In contrast, chronic secondary musculoskeletal pain refers to pain attributable to a known medical condition—such as cancer, a surgical lesion, or a specific musculoskeletal disorder—and corresponds to what has traditionally been considered chronic pain [5,19]. Given that CPMP is frequently associated with substantial emotional and cognitive symptoms, its clear diagnostic delineation reinforces the need to investigate this condition as a distinct entity within a biopsychosocial framework [20].

Thus, due to the lack of research based on the diagnostic subtypes established in the ICD-11, it is essential to determine the implications of chronic primary musculoskeletal pain (CPMP) in terms of psychological distress and sleep quality [5]. In this context, the objectives of the present study were (1) to assess psychopathological symptomatology and sleep quality in individuals diagnosed with CPMP and in healthy controls; (2) to examine group differences in psychological distress and sleep quality between individuals with and without a CPMP diagnosis; (3) to analyse the relationship between pain intensity, psychopathological symptomatology, and sleep quality in patients with CPMP; and (4) to explore whether pain intensity may play a meaningful role in psychological distress and poor sleep quality in this population.

## 2. Materials and Methods

### 2.1. Participants

This is an observational study with a cross-sectional design. Using a non-probabilistic sampling method, we recruited a total of 30 patients diagnosed with chronic primary musculoskeletal pain (CPMP), following the criteria established in the International Classification of Diseases (ICD-11) [5]. To participate in the study, patients had to be between 18 and 80 years old. This age range was selected based on epidemiological evidence indicating that chronic musculoskeletal pain can occur in younger adults and becomes increasingly prevalent and persistent with age, particularly from mid-adulthood up to around the age of 80, before the prevalence of multimorbidity and functional disability becomes substantially higher [21,22]. Patients were selected from a centre specialising in the comprehensive rehabilitation of individuals with physical and neurological conditions, registered with the Regional Health Service of Extremadura, Spain. All patients had a prior clinical diagnosis of CPMP, documented in their medical records. As part of the initial assessment, the diagnosis was confirmed using a structured interview, ensuring that the pain had persisted for at least three months and that no underlying medical, neurological, or psychiatric condition could better account for the symptoms, in accordance with the ICD-11 criteria distinguishing chronic primary from chronic secondary musculoskeletal pain. Based on these criteria, patients were excluded if they (1) had been experiencing pain for less than three months, (2) had a medical or psychiatric condition, injury, or accident that could better explain their symptoms, (3) were taking psychotropic or neurological medications (e.g., antidepressants, antipsychotics, anticonvulsants, or benzodiazepines), whether for the treatment of chronic pain or for other unrelated conditions, or (4) had received any physical or psychological intervention prior to the study.

Additionally, a control group of 30 healthy individuals was recruited to serve as a baseline for comparison. These participants, also aged between 18 and 80, had no history of chronic pain or psychiatric conditions and were not receiving psychological interventions during the study period. They were recruited through community advertisements in local centres and online platforms.

### 2.2. Instruments

1.Sociodemographic data: Basic participant information was collected using a structured questionnaire. Specifically, this instrument gathered data on age, sex, and civil status for all participants. Additionally, for the CPMP group, information was recorded on pain location, pain duration (in years), and the use of prescribed medication for pain management.2.The Numeric Pain Rating Scale (NPRS) [23]: This is a self-reporting tool for assessing pain intensity. This scale has been widely used in research into chronic pain, including CMP [24,25]. Participants were instructed to rate their pain intensity over the last 24 h using an 11-point numerical scale, where 0 represents “no pain” and 10 represents “the worst pain imaginable”. The commonly used ranges for pain intensity classification are 0 (no pain), 1–3 (mild pain), 4–6 (moderate pain), and 7 or higher (severe pain) [23,25]. The NPRS has demonstrated strong test–retest reliability, with intraclass correlation coefficients (ICC) ranging from 0.58 to 0.96 in patients with a diagnosis compatible with CPMP [26,27,28].3.The Symptom Checklist-90-R (SCL-90-R) [29]: This is a self-report instrument that assesses psychological distress. It consists of 90 items with a Likert-type response scale ranging from 0 (not at all) to 4 (extremely). The items are grouped into nine symptom dimensions: somatisation, obsessive–compulsive, interpersonal sensitivity, depression, anxiety, hostility, phobic anxiety, paranoid ideation, and psychoticism. Additionally, the scale includes three global distress indices: the Global Severity Index (GSI), which reflects overall psychological distress; the Positive Symptom Total (PST), which indicates the number of reported symptoms regardless of intensity; and the Positive Symptom Distress Index (PSDI), which assesses the average intensity of the reported symptoms. According to Derogatis [29], T-scores between 40 and 60 fall within the normative range, while scores ≥60 indicate clinically significant psychological distress. Regarding reliability, Cronbach’s alpha for this instrument is approximately α = 0.80 in general and clinical populations. In this study, internal consistency was α = 0.79.4.The Pittsburgh Sleep Quality Index (PSQI) [30]: This is a self-report instrument that assesses overall sleep quality. The scale consists of 19 questions with a Likert-type response scale ranging from 0 to 3, answered by the participant. Additionally, five optional questions for a roommate provide qualitative insights but do not contribute to the final score. The self-reported items are grouped into seven components: subjective sleep quality, sleep latency, sleep duration, sleep efficiency, sleep disturbances, the use of sleep medication, and daytime dysfunction. The total score ranges from 0 to 21, with higher scores indicating worse sleep quality. According to Buysse et al. [30], a total score below 5 indicates “good sleep quality”, while scores ≥5 suggest “poor sleep quality”. Regarding reliability, studies report Cronbach’s alpha between α = 0.64 and α = 0.83 in general and clinical populations [30,31]. In this study, internal consistency was α = 0.75.

### 2.3. Procedure

The research was conducted in a clinic specialising in the comprehensive rehabilitation and recovery of individuals with physical and neurological conditions. All participants provided informed consent prior to inclusion in the study. The objectives of the research were explained to them, and both the anonymity and confidentiality of their responses were ensured. The instruments were administered individually, with each evaluation session lasting approximately 45 min. To ensure the reliability and validity of the collected data, all evaluators received prior training on the administration of the instruments. Evaluators were present throughout the entire process to address any questions and to ensure that the tests were completed appropriately. No issues arose during the assessments.

All procedures followed the ethical standards of the University of Extremadura (Ref.: 157/2023), as well as the principles of the 1964 Declaration of Helsinki and its subsequent amendments or comparable ethical standards. The study complied with the requirements of Organic Law 3/2018 of December 5th, on the Protection of Personal Data and Guarantee of Digital Rights, ensuring both data confidentiality and secure storage.

### 2.4. Data Analysis

This observational study followed a cross-sectional design. First, a descriptive analysis was conducted for sociodemographic and pain-related variables, psychopathological symptomatology, and sleep quality, using measures of central tendency, variability parameters, and frequency distribution analysis. Normality of continuous variables was assessed using the Shapiro–Wilk test. Secondly, given the non-parametric nature of the data, the Mann–Whitney U test was used to compare the CPMP group and the healthy control (HC) group. Categorical variables were expressed as numbers (%) and compared using Pearson’s χ^2^ test with Yates’ correction or Fisher’s exact test, as appropriate. Thirdly, to examine the relationship between pain intensity, psychopathological symptomatology, and sleep quality, partial correlation analyses were conducted, controlling for sex, age, and pain duration. Finally, hierarchical linear regression was conducted within the CPMP group to examine whether pain intensity may play a meaningful role in psychopathological symptomatology and sleep quality after controlling for sex, age, and pain duration. The covariates were entered in the first step of the model, and pain intensity was entered in the second step. To control for multiple comparisons and reduce the risk of Type I errors, the Benjamini–Hochberg correction (FDR = 0.05) was applied to all statistical analyses. This procedure controls the false discovery rate, allowing for fewer false positives among significant results while maintaining statistical power. Statistical significance was set at *p* < 0.05, and all analyses were conducted using SPSS version 27 for Windows.

To justify the sample size, an a priori power calculation was performed assuming a medium-to-large effect size (Cohen’s d = 0.65), a significance level of α = 0.05, and a desired power of 0.80. The analysis indicated that a total sample of 60 participants (30 per group) would be adequate. This estimation was also supported by previous studies examining similar relationships between chronic pain and psychopathological symptoms, which used comparable sample sizes [32,33,34]. To evaluate the statistical power of our analyses, we also conducted a post hoc power analysis. For psychological distress (SCL-90-R GSI), the comparison between chronic pain patients and control participants yielded a large effect size (d = 0.74) with high statistical power (1 − β = 0.78). Similarly, for sleep quality (PSQI), the effect size was also large (d = 0.61), with moderate power (1 − β = 0.62). These results indicate that the study had sufficient power to detect the observed effects.

## 3. Results

The study included a total of 60 participants, with 30 in the CPMP group and 30 in the HC group. The sociodemographic characteristics of the participants are presented in Table 1. Participants ranged in age from 41 to 75 years, with a mean age of 57.60 (SD = 8.51; range = 41–75) in the CPMP group and 54.13 (SD = 5.82; range = 44–69) in the HC group; no statistically significant differences were found between groups. Regarding sex, 26.7% of participants in the CPMP group were male (*n* = 8) and 73.3% were female (*n* = 22), whereas, in the HC group, 60% were male (*n* = 18) and 40% were female (*n* = 12), revealing a statistically significant difference. No significant differences were observed in marital or occupational status.

Additional clinical information was collected from the CPMP group. All patients reported low back pain, with an average pain duration of 11.95 years (SD = 1.85; range = 1–40). Regarding pain intensity, the chronic pain group reported moderate levels on the NPRS (M = 5.23, SD = 1.63). Additionally, 86.7% (*n* = 26) of participants were taking prescribed pain medication, the most commonly used being paracetamol (31%), enantyum (24%), ibuprofen (17%), aceclofenac (14%), and diclofenac (14%) (Table 1).

Results of the Mann–Whitney U test for SCL-90-R and PSQI scores between the chronic pain and control groups are presented in Table 2. The only SCL-90-R dimension that did not show a statistically significant difference between groups was paranoid ideation. All other dimensions and global indices revealed statistically significant differences, with the CPMP group scoring higher on all remaining scales. Within the CPMP group, somatisation reached clinical significance, and obsessive–compulsive symptoms approached that threshold. The remaining dimensions fell within normative or below-average ranges. Scores on the GSI and PST were close to clinical significance, while PSDI scores were normative. The HC group scored below the normative range across all dimensions and indices, indicating no signs of psychological distress. Regarding the PSQI, significant group differences were observed in global scores, with the CPMP group reporting poorer sleep quality than the HC group. The CPMP group was classified as having moderately impaired sleep quality, whereas the HC group fell within the mildly impaired range.

Table 3 presents the partial correlation analyses examining the relationship between pain intensity, psychological symptoms, and sleep quality in the CPMP group. Specifically, significant positive correlations were found between pain intensity and the dimensions of somatisation, obsessive–compulsive symptoms, interpersonal sensitivity, depression, anxiety, hostility, phobic anxiety, paranoid ideation, and psychoticism, as well as with the global indices GSI, PST, and PSDI. Pain intensity was also positively correlated with poorer sleep quality, as measured by the PSQI. These findings indicate that, in individuals with CPMP, greater pain intensity is associated with more severe psychological distress and reduced sleep quality.

Finally, to explore the potential role of pain intensity in psychological distress and sleep quality in the CPMP group, a hierarchical linear regression analysis was conducted (Table 4). Pain intensity significantly accounted for variance in all SCL-90-R dimensions, the global indices (GSI, PST, and PSDI), and the global PSQI score, even after controlling for age, sex, and pain duration. These findings suggest that higher pain intensity may play a meaningful role in greater psychological distress and reduced sleep quality in individuals with CPMP.

## 4. Discussion

Given the growing body of evidence on chronic primary musculoskeletal pain (CPMP) since the official adoption of the ICD-11 classification in 2019, it has become increasingly important to explore the psychological and functional consequences associated with this diagnosis [3,5]. This study aimed to examine psychopathological symptomatology and sleep quality in individuals with CPMP, and to determine whether pain intensity is related to psychological distress and poor sleep quality in this population.

Based on the results, we found that patients with CPMP exhibited a markedly more severe psychopathological profile compared to healthy controls. Specifically, they presented moderate levels of somatisation, obsessive–compulsive traits, interpersonal sensitivity, depression, anxiety, hostility, psychoticism, and, to a lesser extent, phobic anxiety and paranoid ideation. These symptoms were reflected in significantly higher scores on the global distress indices of the SCL-90-R, including the Global Severity Index (GSI), the Positive Symptom Total (PST), and the Positive Symptom Distress Index (PSDI). Furthermore, sleep quality was significantly poorer in the CPMP group, suggesting persistent difficulties in areas such as sleep duration, latency, and maintenance. In contrast, the healthy control group reported minimal psychological symptoms and relatively better sleep quality. These findings are consistent with the previous literature, indicating that individuals with chronic low back pain experience a wide spectrum of psychological difficulties beyond depression and anxiety, including somatic symptoms, emotional distress, and impaired sleep [35,36,37,38]. Such manifestations may be linked to maladaptive pain coping strategies, a limited acceptance of chronic pain, beliefs about one’s lack of control, avoidance behaviours, and reduced overall well-being [24,39,40]. In line with previous studies comparing clinical and non-clinical samples [34,41,42,43], the present results emphasise the importance of addressing both psychological and somatic dimensions in the assessment and treatment of chronic pain.

Regarding the relationship between pain intensity, psychological distress, and sleep quality in the CPMP group, our results showed that higher pain intensity was significantly associated with greater psychopathological symptomatology across all SCL-90-R dimensions, as well as with poorer sleep quality. These findings suggest a robust association between physical suffering and emotional disturbances, reinforcing the notion that more intense pain is linked to a greater psychological burden and disrupted sleep. The particularly high correlations between pain intensity and global indices such as the GSI may reflect this cumulative burden, as highlighted in recent frameworks that define CPMP as a condition with significant psychopathological implications [20]. Moreover, regression analyses confirmed that pain intensity may play a meaningful role in both psychological distress and sleep quality, even after adjusting for relevant covariates. These results are consistent with previous studies reporting moderate-to-severe pain in individuals with chronic low back pain (CLBP) [33,36,38] and with evidence showing that pain chronification is accompanied by increased depressive symptoms [44,45,46]. Although some studies have reported mixed findings regarding the predictive value of pain intensity [47,48], our findings suggest that pain intensity potentially contributes to both psychological and sleep-related outcomes in CPMP. In line with our results, several studies have also found that greater pain intensity contributes to poorer sleep quality in patients with chronic low back pain [8,49,50,51], suggesting that pain severity may contribute to the development or persistence of sleep disturbances in this population. Taken together, these findings underscore the importance of assessing pain intensity not only as a physical symptom, but also as a potential indicator of broader psychological vulnerability in chronic pain management.

This research has enabled us to identify several areas of vulnerability in individuals with CPMP, contributing to a deeper understanding of the psychopathological profile associated with this condition and its relationship with pain intensity. Nonetheless, the study presents some limitations. Due to its cross-sectional design, it was not possible to examine the progression of psychopathological symptomatology and sleep quality over time. Additionally, given the characteristics of the target population and the difficulty in identifying cases of chronic musculoskeletal pain that met the ICD-11 criteria, which require the pain to not be attributable to a known medical condition, a non-probabilistic sampling method was employed. This may have introduced selection bias and limited the representativeness of the sample, as all participants in the clinical group had low back pain. As a result, the findings primarily reflect this specific subtype of chronic primary musculoskeletal pain and may not be generalizable to other presentations within this diagnostic category. Moreover, despite the use of screening protocols and statistical adjustments, differences in recruitment strategy may also have affected the comparability between the CPMP and control groups, since participants were recruited through different sources. The relatively small sample size is another limitation, potentially affecting the generalizability of the results and limiting the statistical power for conducting subgroup analyses, such as comparisons across age groups, which were not performed. The sex imbalance between groups also warrants consideration, as the CPMP group included significantly more women than the control group. Given that women tend to report higher levels of psychological distress, this difference may have contributed to the group differences observed in related measures. Although sex was included as a covariate in the analyses, this statistical adjustment may not have fully accounted for the influence of group composition, which should be taken into account when interpreting the findings. Finally, the assessment of psychopathological symptomatology, sleep quality, and pain intensity relied on self-reported measures, which may be susceptible to response biases and reflect subjective experiences that are influenced by individual interpretations or emotional states, potentially compromising data reliability. To address this limitation, future studies should consider incorporating objective methods such as actigraphy for sleep or clinician-rated assessments for psychopathology and pain in order to improve the validity of the findings and complement self-reported data with more independent sources of information, thereby providing a more comprehensive understanding of these conditions [52].

For future research, it would be advisable to conduct longitudinal studies beginning either prior to the onset of the condition or at the early stages of pain, in order to identify predictive factors related to increases in pain intensity, psychopathological symptomatology, and/or sleep quality deficits [9,35]. One of the most rapidly emerging perspectives highlights the existence of bidirectional relationships between chronic pain, psychological distress, and sleep disturbances. This approach suggests that chronic pain may increase the likelihood of developing maladaptive emotions, thoughts, and behaviours related to pain—and vice versa. Therefore, it would be important to consider the bidirectional nature of these variables in future studies [12,13,14]. Identifying early predictive factors for pain onset could, in turn, allow for the development of early interventions aimed at reducing the risk of pain emergence or minimising the likelihood of chronification. In addition, future research should consider including measures of health-related quality of life, as these can capture the broader impact of chronic pain on daily functioning, social participation, and overall well-being, thereby complementing the psychological and sleep-related variables examined in the present study and contributing to a more comprehensive understanding of CPMP within the biopsychosocial model.

Despite the above considerations, a key strength of this study lies in being among the first to evaluate the psychopathological profile of patients with CPMP and to compare it with that of healthy controls without any diagnosis, thereby extending previously reported findings in chronic low back pain populations. Furthermore, it offers new evidence regarding the associations between pain intensity and both psychopathological symptomatology and sleep quality in this population, as well as the potential role of pain intensity in these domains. Another methodological strength is that the statistical analyses included age, sex, and pain duration as covariates, allowing us to partially account for potential confounding factors. The findings from this study provide a valuable starting point for addressing CPMP from a therapeutic perspective, opening the door to the development of specific psychological interventions tailored to the identified deficits.

## 5. Conclusions

In conclusion, this study has contributed to identifying the psychopathological profile of individuals with CPMP. The findings indicate that patients with CPMP exhibit greater psychopathological symptomatology and poorer sleep quality compared to healthy controls without a diagnosis. Moreover, higher pain intensity in the CPMP group was associated with increased psychological distress and greater sleep disturbances. Pain intensity also emerged as a potential contributor to both heightened psychopathological symptoms and more pronounced deficits in sleep quality. Further research is needed to replicate these findings in larger and more diverse samples, and to explore the potential benefits of integrated psychological interventions in this population.

## Figures and Tables

**Table 1 healthcare-13-01965-t001:** Sociodemographic characteristics of the participants.

	CPMP (*n* = 30)	HC (*n* = 30)	*p*
Age (average in years ± SD)	57.60 ± 8.51	54.13 ± 5.82	0.087 ^1^
Range (years)	41–75	44–69	
Sex, *n* (%)			
Male	8 (26.7)	18 (60)	0.019 ^2^
Female	22 (73.3)	12 (40)	
Civil status, *n* (%)			
Single	2 (6.7)	3 (10)	1 ^3^
Married	24 (80)	24 (80)	
Divorced	2 (6.7)	2 (6.7)	
Widowed	2 (6.7)	1 (3.3)	
Occupational status, *n* (%)			
Employed	20 (66.7)	24 (80)	0.164 ^3^
Unemployed	4 (13.3)	5 (16.7)	
Retired	6 (20)	1 (3.3)	
Site of pain, *n* (%)			
Lumbar	30 (100)	NA	NA
Cervical	0	NA	NA
Thorax	0	NA	NA
Other	0	NA	NA
Duration of pain (average in years ± SD)	11.95 ± 1.85	NA	NA
Range (years)	1–40	NA	NA
NPRS (mean ± SD)	5.23 ± 1.63	NA	NA
Medication for pain, *n* (%)			
Yes	26 (86.7)	NA	NA
No	4 (13.3)	NA	NA

Note. CPMP: chronic primary musculoskeletal pain; HC: healthy control; NPRS: The Numeric Pain Rating Scale; NA: Not applicable. Values are presented as mean ± standard deviation or number (%). ^1^
*p*-values are generated from Mann–Whitney U test. ^2^
*p*-values are generated from Pearson’s chi-squared test. ^3^
*p*-values are generated from Fisher’s exact test.

**Table 2 healthcare-13-01965-t002:** Comparison between the CPMP group and the HC group in the SCL-90-R and PSQI tests.

	CPMP (*n* = 30)	HC (*n* = 30)	*Z*	Adjusted *p*-Value (Benjamini–Hochberg)
SCL-90-R T scores				
1. SOM	60.93 ± 6.50	45.93 ± 8.79	−5.56	0.001 ^1^
2. O-C	56.83 ± 10.21	43.20 ± 8.22	−4.89	0.001 ^1^
3. I-S	51.23 ± 10.30	41.43 ± 7.30	−3.82	0.001 ^1^
4. DEP	54.40 ± 10.05	39.53 ± 5.94	−5.33	0.001 ^1^
5. ANX	50.53 ± 7.92	41.20 ± 8.47	−4.67	0.001 ^1^
6. HOS	51.10 ± 8.30	41.80 ± 8.50	−3.76	0.001 ^1^
7. PHOB	47.23 ± 13.97	39.20 ± 8.32	−2.50	0.013 ^1^
8. PAR	46.73 ± 11.59	42.23 ± 8.38	−1.29	0.197 ^1^
9. PSY	51.40 ± 14.10	39.70 ± 10.56	−3.09	0.002 ^1^
GSI	57.27 ± 9.53	40.97 ± 6.69	−5.73	0.001 ^1^
PST	59.13 ± 10.06	43.83 ± 8.03	−5.11	0.001 ^1^
PSDI	50.03 ± 10.52	38.23 ± 5.69	−4.62	0.001 ^1^
PSQI—Global score	11.17 ± 3.90	5.80 ± 3.32	−4.73	0.001 ^1^
Good sleep quality	2 (6.7)	18 (60)	NA	0.001 ^2^
Bad sleep quality	28 (93.3)	12 (40)		

Note. The Symptom Checklist-90-R; SOM: somatisation; O-C: obsessive–compulsive; I-S: interpersonal sensitivity; DEP: depression; ANX: anxiety; HOS: hostility; PHOB: phobic anxiety; PAR: paranoid ideation; PSY: psychoticism; GSI: Global Severity Index; PST: Positive Symptom Total; PSDI: Positive Symptom Distress Index; PSQI: The Pittsburgh Sleep Quality Index; CPMP: chronic primary musculoskeletal pain; HC: healthy control; NA: Not applicable. Values are presented as mean ± standard deviation or number (%). ^1^
*p*-values are generated from Mann–Whitney U test. ^2^
*p*-values are generated from Pearson’s chi-squared test.

**Table 3 healthcare-13-01965-t003:** Partial correlation analysis between pain intensity, psychopathological symptomatology, and sleep quality in the CPMP group, controlling for age, sex, and pain duration.

	CPMP (*n* = 30)
	NPRS
	Correlation Coefficient (r)
SCL-90-R T Scores	
1. SOM	0.753 **
2. O-C	0.542 **
3. I-S	0.678 **
4. DEP	0.823 **
5. ANX	0.800 **
6. HOS	0.586 **
7. PHOB	0.474 *
8. PAR	0.362 *
9. PSY	0.641 **
GSI	0.838 **
PST	0.735 **
PSDI	0.574 **
PSQI—Global Score	0.785 **

Note. NPRS: The Numeric Pain Rating Scale; SCL-90-R: The Symptom Checklist-90-R; SOM: somatisation; O-C: obsessive–compulsive; I-S: interpersonal sensitivity; DEP: depression; ANX: anxiety; HOS: hostility; PHOB: phobic anxiety; PAR: paranoid ideation; PSY: psychoticism; GSI: Global Severity Index; PST: Positive Symptom Total; PSDI: Positive Symptom Distress Index; PSQI: The Pittsburgh Sleep Quality Index; CPMP: chronic primary musculoskeletal pain; *p*-values adjusted using the Benjamini–Hochberg procedure: * *p* < 0.05; ** *p* < 0.01.

**Table 4 healthcare-13-01965-t004:** Hierarchical linear regression analysis examining the potential role of pain intensity in psychopathological symptomatology and sleep quality in the CPMP group, controlling for age, sex, and pain duration.

	CPMP (*n* = 30)			
	NPRS			
	Adjusted R^2^	β	t	Adjusted *p*-Value (Benjamini–Hochberg)
SCL-90-R T Scores				
1. SOM	0.42	0.68	4.15	0.013
2. O-C	0.35	0.61	3.51	0.003
3. I-S	0.56	0.76	5.32	0.006
4. DEP	0.59	0.75	5.44	0.004
5. ANX	0.64	0.80	6.15	0.003
6. HOS	0.56	0.74	5.19	0.003
7. PHOB	0.22	0.48	2.52	0.019
8. PAR	0.33	0.52	2.95	0.008
9. PSY	0.53	0.69	4.63	0.002
GSI	0.68	0.87	7.08	0.002
PST	0.67	0.84	6.73	0.002
PSDI	0.27	0.53	2.89	0.009
PSQI—Global Score	0.55	0.74	5.10	0.001

Note. NPRS: The Numeric Pain Rating Scale; SCL-90-R: The Symptom Checklist-90-R; SOM: somatisation; O-C: obsessive–compulsive; I-S: interpersonal sensitivity; DEP: depression; ANX: anxiety; HOS: hostility; PHOB: phobic anxiety; PAR: paranoid ideation; PSY: psychoticism; GSI: Global Severity Index; PST: Positive Symptom Total; PSDI: Positive Symptom Distress Index; PSQI: The Pittsburgh Sleep Quality Index.

## Data Availability

The authors confirm that all data generated or analysed during this study are included in this published article.

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
