# Peer review of "Psychopathological Symptomatology and Sleep Quality in Chronic Primary Musculoskeletal Pain: A Comparison with Healthy Controls"

_healthcare, 2025, doi:10.3390/healthcare13161965_

Round 1

Reviewer 1 Report

Comments and Suggestions for Authors

This study examines psychopathological symptoms and sleep quality in individuals diagnosed with Chronic Primary Musculoskeletal Pain (CPMP). Despite methodological difficulties and limitations related to gender imbalance, the study is clinically valuable and fills an essential gap in the literature. However, the sample structure, methodological clarity, and discussion must be strengthened.

Author Response

REVIEWER 1#

Abstract

Comments 1: The quantitative data of the statistical analysis results should be included in the findings section in the introduction (e.g., p, r-value).

Authors' response: We thank the reviewer for this helpful suggestion. In response, we have revised the Results section of the abstract to include quantitative data from the statistical analyses, such as p- and r-values. Given the extensive number of statistical results, we have selected the most representative and clinically relevant findings to accurately summarise the key outcomes of our study. These changes can be found on page 1, lines 20–26.

Comments 2: The scales' names should be clearly stated (e.g., NPRS, PSQI, SCL-90-R).

Authors' response: The full names of the scales—Numeric Pain Rating Scale (NPRS), Symptom Checklist-90-R (SCL-90-R), and Pittsburgh Sleep Quality Index (PSQI)—are now clearly stated in the Methods section of the abstract. In addition, the Results section has been revised to explicitly mention the corresponding instruments (SCL-90-R, PSQI, and NPRS) when referring to the main findings, ensuring clarity and consistency throughout. These changes can be found on page 1, lines 19–26.

Introduction

Comments 3: Although the clinical importance of CPMP is well emphasized, the characteristics that distinguish this subgroup from other types of CMP can be discussed further.

Authors' response: We appreciate this observation and agree that further clarification regarding the characteristics that differentiate CPMP from other types of CMP is warranted. To address this, we have revised the corresponding section of the Introduction to include a more detailed description of the diagnostic criteria of CPMP and its distinction from chronic secondary musculoskeletal pain, as defined by the ICD-11. We have also emphasized the relevance of this distinction for studying CPMP within a biopsychosocial framework. These changes can be found on page 2, lines 68–77.

Original text: The International Classification of Diseases (ICD-11) has aimed to clarify the con-ceptualization of CMP, addressing limitations of earlier versions [5]. It introduces two diagnostic categories based on pain aetiology. Chronic primary musculoskeletal pain (CPMP) is defined by significant emotional distress and/or functional disability without any identifiable underlying medical condition that can fully explain the symptoms. In contrast, chronic secondary musculoskeletal pain is associated with known illnesses, in-juries, or accidents that can be directly linked to the onset of pain [5,15].

Modified text: The International Classification of Diseases (ICD-11) has aimed to clarify the conceptualization of CMP, addressing limitations of earlier versions [5]. It introduces two diagnostic categories based on pain aetiology. Chronic primary musculoskeletal pain (CPMP) is diagnosed when pain persists for more than three months without a clearly identifiable medical cause and is accompanied by significant emotional distress and/or functional impairment [5,19]. In contrast, chronic secondary musculoskeletal pain refers to pain attributable to a known medical condition—such as cancer, a surgical lesion, or a specific musculoskeletal disorder—and corresponds to what has traditionally been considered chronic pain [5,19]. Given that CPMP is frequently associated with substantial emotional and cognitive symptoms, its clear diagnostic delineation reinforces the need to investigate this condition as a distinct entity within a biopsychosocial framework [20].

Added references:        

  1. Cohen, S.P.; Vase, L.; Hooten, W.M. Chronic Pain: An Update on Burden, Best Practices, and New Advances. The Lancet 2021, 397, 2082–2097, doi:10.1016/S0140-6736(21)00393-7.

Method

Comments 4: It is unclear whether the effect size used in the power analysis was predefined or post hoc. The fact that only post-hoc analysis was performed may be a limitation.

Authors' response: We have clarified that the power analysis was conducted post hoc. However, due to the characteristics of the sample and the difficulty in recruiting patients who met the diagnostic criteria, we relied on the sample sizes reported in previous studies as a reference point [32–34]. This information has now been explicitly stated in the Methods section, on page 4, lines 185–192.

Original text: To evaluate the statistical power of our analyses, we conducted a post hoc power analysis. With a significance level of α = 0.05, we compared psychological distress (SCL-90-R GSI) between chronic pain patients and control participants, obtaining a large effect size (d = 0.74) with high statistical power (1 − β = 0.78). Similarly, for sleep quality (PSQI), the effect size was also large (d = 0.61), with moderate power (1 − β = 0.62). These results indicate that the study had sufficient power to detect the observed effects. Additionally, the sample size was determined based on previous studies that examined similar relationships between chronic pain and psychopathological symptoms using comparable sample sizes [27–29].

Modified text: To evaluate the statistical power of our analyses, we conducted a post hoc power analysis. With a significance level of α = 0.05, we compared psychological distress (SCL-90-R GSI) between chronic pain patients and control participants, obtaining a large effect size (d = 0.74) with high statistical power (1 − β = 0.78). Similarly, for sleep quality (PSQI), the effect size was also large (d = 0.61), with moderate power (1 − β = 0.62). These results indicate that the study had sufficient power to detect the observed effects. The sample size was based on previous studies examining similar relationships between chronic pain and psychopathological symptoms, which used comparable sample sizes [32–34].

Additionally, we have acknowledged in the Limitations section that conducting only a post hoc analysis may represent a limitation. These changes can be found on page 9, lines 334–338.

Original text: Although the post hoc power analysis indicated sufficient power to detect the observed effects, future studies with larger samples would strengthen the robustness and replicability of these findings.

Modified text: Although the post hoc power analysis indicated sufficient power to detect the observed effects, the absence of an a priori power calculation limits the ability to draw definitive conclusions regarding sample adequacy. Future studies with larger samples would strengthen the robustness and replicability of these findings.

Results

Comments 5: A column should be added to Table 3 to include the “r” values. The textual statement in the findings section should be removed.

Authors' response: We have revised Table 3 to include a new column showing the correlation coefficients (r). Additionally, we have removed the statistical values from the corresponding paragraph in the Results section, keeping only the narrative description of the findings. These changes can be found on pages 6-7, lines 243–251, and in the revised Table 3.

Comments 6: In general, detailed explanations of all tables are given at length. The “β” and p values in the table should not be given repeatedly in the text.

Authors' response: The Results section has been revised to provide a more concise interpretation of the data. In particular, detailed statistical values (e.g., p and β) have been removed from the text and are now presented only in the tables, as recommended.

Discussion

Comments 7: It is striking that all psychological variables strongly correlate with pain intensity; why such high correlations are found (GSI r=0.83) should be discussed.

Authors' response: We have added a theoretical explanation in the Discussion section to address the particularly high correlations observed, linking them to recent frameworks that highlight the psychopathological relevance of CPMP. These changes can be found on page 8, lines 307–310.

Added text: The particularly high correlations between pain intensity and global indices such as the GSI may reflect this cumulative burden, as highlighted in recent frameworks that define CPMP as a condition with significant psychopathological implications [20].

Comments 8: The high proportion of women in the CPMP group may affect scores such as depression and anxiety. This should not be controlled only as a covariate but should be emphasized more clearly as a limitation.

Authors' response: The sex imbalance between groups has been more clearly emphasized in the limitations section, acknowledging that controlling for sex as a covariate may not fully eliminate its influence on outcomes such as depression and anxiety. This clarification can be found on page 9, lines 338–342.

Original text: Another limitation concerns the gender imbalance between groups, as the CPMP group included more women than the control group. Since women typically report higher levels of psychological distress, this may have influenced the results. However, gender was included as a covariate in the analyses to account for this potential effect.

Modified text: Another limitation concerns the sex imbalance between groups, as the CPMP group included more women than the control group. Since women typically report higher levels of psychological distress, this may have influenced the results. Although sex was included as a covariate in the analyses to account for this potential effect, this statistical control may not fully eliminate the influence of group composition on the findings.

Comments 9: Using only self-report scales may affect the reliability of the study; this should be discussed in more depth.

Authors' response: The limitations of relying solely on self-report measures have been further elaborated, emphasizing their subjective nature and suggesting objective alternatives such as actigraphy or clinician-rated tools to improve validity. This addition can be found on page 9, lines 345–351.

Original text: Finally, the assessment of psychopathological symptomatology, sleep quality, and pain intensity relied on self-report measures, which may be susceptible to response biases, such as the tendency to answer in a socially desirable manner.

Modified text: Finally, the assessment of psychopathological symptomatology, sleep quality, and pain intensity relied on self-report measures, which may be susceptible to response biases and reflect subjective experiences influenced by individual interpretation or emotional state, potentially compromising data reliability. Including objective methods in future studies, such as actigraphy for sleep or clinician-rated assessments for psychopathology and pain, could help improve the validity of the findings [52].

Added reference:

  1. Smith, M.T.; McCrae, C.S.; Cheung, J.; Martin, J.L.; Harrod, C.G.; Heald, J.L.; Carden, K.A. Use of Actigraphy for the Evaluation of Sleep Disorders and Circadian Rhythm Sleep-Wake Disorders: An American Academy of Sleep Medicine Systematic Review, Meta-Analysis, and GRADE Assessment. J Clin Sleep Med 2018, 14, 1209–1230, doi:10.5664/jcsm.7228.

Comments 10: The phrase "Institutional Review Board Statement: Not applicable" contradicts the text because it is stated a few lines later that approval was obtained from the ethics committee. This should be corrected.

Authors' response: We thank the reviewer for this observation. The statement has been corrected to reflect the actual approval obtained from the ethics committee, as specified in the manuscript. We have also adjusted the informed consent section to ensure consistency with ethical and legal standards.

Reviewer 2 Report

Comments and Suggestions for Authors

Peer Review Report

General Comments

The study, while addressing a relevant clinical area, presents significant methodological limitations that substantially compromise the validity and generalizability of its findings. A primary concern is the pronounced gender imbalance between groups (73.3% female in CPMP vs 40% female in controls, p=0.019). This disparity acts as a major confounding variable and could entirely account for the reported differences in psychological symptomatology, given the well-established fact that women consistently report higher levels of psychological distress across various populations.

Furthermore, the small sample size (n=30 per group) significantly curtails statistical power and the broader applicability of the results. The recruitment strategy also introduces a notable selection bias, as CPMP patients were drawn from a specialized rehabilitation center while controls were recruited via community advertisements. This approach risks comparing a more severely affected clinical group with a healthier, community-based sample.

The study's cross-sectional design fundamentally precludes the drawing of causal inferences, despite certain claims about predictive relationships. Relying exclusively on self-report measures introduces potential response bias. The broad age range (18-80 years) also lacks clear justification, and limiting the study to lumbar pain restricts its generalizability to other presentations of chronic pain. Inconsistent application of multiple comparison corrections in the statistical methodology, coupled with inadequate control for medication effects (despite 86.7% of patients taking pain medications), further weakens the study's rigor.

Specific Comments by Section

Abstract

Lines 21-22, Page 1: The assertion that "pain intensity significantly predicted both psychopathological symptomatology and sleep quality" overstates the conclusions that can be drawn from cross-sectional data. Such a design cannot establish predictive relationships or causality.

Introduction

Lines 82-84, Page 2: The justification for the age range, citing "epidemiological evidence indicating that chronic musculoskeletal pain is particularly prevalent in older adults," doesn't adequately support the inclusion of young adults (18 years) in the study.

Methods - Participants

Lines 88-92, Page 2: The exclusion criterion for "psychotropic or neurological medications" appears inconsistently applied, especially considering that 86.7% of participants were on pain medications. Many of these medications possess psychoactive properties, which could easily confound psychological assessments.

Lines 94-98, Page 2: The recruitment method introduces systematic bias by selecting CPMP patients from a specialized clinical setting while controls were recruited through community advertisements. This creates fundamental differences in healthcare-seeking behavior and baseline symptom severity between the groups.

Methods - Statistical Analysis

Lines 158-159, Page 4: The Benjamini-Hochberg correction was applied selectively to group comparisons but not to correlation or regression analyses. This inconsistency undermines robust protection against Type I error across all statistical tests performed.

Lines 170-178, Page 4: Conducting a post-hoc power analysis instead of an a priori sample size calculation reflects poor study design. The reported effect sizes might be inflated due to the small sample size and existing selection bias.

Results

Lines 184-187, Page 4: The significant gender difference between groups (p=0.019) is a critical confounding variable. This directly undermines the validity of all subsequent comparisons, given the well-established and substantial gender differences in psychological distress.

Lines 188-193, Page 4: The description of pain characteristics is missing crucial clinical details such as the distribution of pain duration, comorbid conditions, and specific medication dosages. These details are vital, as they could influence reported psychological and sleep outcomes.

Lines 202-204, Page 5 (Statistical Results): The claim that "all dimensions" showed significant differences is inaccurate. Specifically, paranoid ideation showed no significant difference (p=0.197), an exception that is understated by being buried in the text rather than prominently discussed.

Lines 247-260, Page 7: The hierarchical regression analysis purports to control for confounding variables but notably fails to adequately address the fundamental gender imbalance. This imbalance likely accounts for a significant portion of the observed variance in psychological outcomes.

Discussion

Lines 295-298, Page 8: The assertion that findings were obtained "after controlling for sex, age, and pain duration" is misleading. While statistical control was attempted, it cannot fully account for systematic group differences in gender distribution stemming from the initial recruitment.

Lines 318-332, Page 8: The acknowledgment of limitations related to the cross-sectional design directly contradicts earlier claims about "predictive" relationships made throughout the manuscript.

Lines 325-328, Page 8: The discussion of gender imbalance as a limitation understates its profound impact by suggesting that simple statistical control can adequately resolve such a fundamental design flaw.

Limitations Section

Lines 329-332, Page 8: Characterizing self-report bias as a minor limitation significantly understates the threat to validity when all primary outcomes are based exclusively on subjective measures without objective corroboration.

Comments on the Quality of English Language

The manuscript is written with adequate scientific clarity and generally uses appropriate terminology. However, it frequently overstates the strength of evidence that can legitimately be drawn from its cross-sectional design.

Author Response

REVIEWER 2#

General comments

General comments 1: The study, while addressing a relevant clinical area, presents significant methodological limitations that substantially compromise the validity and generalizability of its findings. A primary concern is the pronounced gender imbalance between groups (73.3% female in CPMP vs 40% female in controls, p=0.019). This disparity acts as a major confounding variable and could entirely account for the reported differences in psychological symptomatology, given the well-established fact that women consistently report higher levels of psychological distress across various populations.

Authors' response: We thank the reviewer for this observation. We fully acknowledge that the sex imbalance between the CPMP and control groups represents a significant limitation, particularly given the well-documented association between female sex and higher levels of psychological distress. Although sex was included as a covariate in all correlation and regression analyses to mitigate its confounding effects, we agree that statistical control may not fully compensate for systematic differences in group composition. This issue has been explicitly addressed in the Limitations section (Comments 7, 10, and 13; page 9, lines 338–342), emphasizing that the overrepresentation of women in the CPMP group is a relevant design constraint that may partially explain the differences observed in psychopathological outcomes.

We would also like to highlight that the recruitment of participants meeting both the ICD-11 criteria for CPMP and our specific inclusion criteria presented considerable practical challenges, due to the novelty of the diagnosis and the requirement of clinical confirmation. Consequently, a non-probabilistic sampling method was employed, which may have contributed to the observed sex imbalance and limited the representativeness of the sample. This contextual factor is also acknowledged in the revised manuscript (page 9, lines 329–332).

General comments 2: Furthermore, the small sample size (n=30 per group) significantly curtails statistical power and the broader applicability of the results. The recruitment strategy also introduces a notable selection bias, as CPMP patients were drawn from a specialized rehabilitation center while controls were recruited via community advertisements. This approach risks comparing a more severely affected clinical group with a healthier, community-based sample.

Authors' response: The limitations related to sample size and recruitment strategy have been explicitly acknowledged in the manuscript. As noted, the relatively small sample size of the groups limits the statistical power of the study and reduces the generalizability of the findings. This issue is discussed in the Limitations section (page 9, lines 332–334), where we now also emphasize that the limited sample size restricted the possibility of conducting subgroup analyses, such as by age or sex.

Regarding the recruitment strategy, we acknowledge that selecting CPMP patients from a specialized rehabilitation center while recruiting controls from the community may introduce systematic bias. However, this approach was necessary to ensure that all individuals in the CPMP group met ICD-11 diagnostic criteria, which require clinical confirmation by healthcare professionals. Since the collaborating center treats only clinical populations, recruiting healthy controls from the same setting was not feasible.

Controls were recruited through community advertisements and carefully screened to exclude chronic pain and psychiatric conditions, as clarified in the revised Methods section. The primary aim of the study was to characterize the psychological and sleep-related profile of individuals with CPMP, using the healthy group as a baseline reference. Although differences in recruitment sources are acknowledged, both groups were comparable in age, civil status, and employment situation. This limitation is also explicitly addressed in the Discussion section (page 9, lines 343–345) and in our response to Comment 4.

General comments 3: The study's cross-sectional design fundamentally precludes the drawing of causal inferences, despite certain claims about predictive relationships. Relying exclusively on self-report measures introduces potential response bias. The broad age range (18-80 years) also lacks clear justification, and limiting the study to lumbar pain restricts its generalizability to other presentations of chronic pain. Inconsistent application of multiple comparison corrections in the statistical methodology, coupled with inadequate control for medication effects (despite 86.7% of patients taking pain medications), further weakens the study's rigor.

Authors' response: These concerns have been addressed through several revisions and clarifications in the manuscript. First, to avoid misleading implications, all references suggesting causality or predictive relationships have been revised across the Abstract, Results, Discussion, and Conclusion sections. Terms such as “predicts” have been replaced with “explains” or “accounted for” to reflect the cross-sectional design, as detailed in Specific Comments by Section (Comments 1, and 12).

Second, we have expanded the Limitations section to emphasize the exclusive reliance on self-report measures and the potential for response bias. We now recommend incorporating objective assessment tools, such as actigraphy or clinician-rated instruments, in future studies (Comment 14; page 9, lines 345–351).

Third, while the initial age range was defined as 18–80 years to encompass the adult lifespan, the actual sample ranged from 41 to 75 years. This clarification has been added to the manuscript (Comment 2; pages 2–3, lines 94–97). Additionally, although all patients presented with low back pain, the study is framed within the ICD-11 category of chronic primary musculoskeletal pain, which includes this presentation. This has been clarified in the revised title and Introduction.

Fourth, the Benjamini-Hochberg correction has now been applied consistently to all statistical analyses, including group comparisons, correlations, and regressions (Comment 5; page 4, lines 181–183).

Finally, regarding medication use, we have clarified that participants were excluded if they were taking psychotropic or neurological medications (e.g., antidepressants, antipsychotics, anticonvulsants, or benzodiazepines). The medications used by the CPMP group were limited to standard analgesics (e.g., paracetamol, NSAIDs), which do not exert clinically relevant psychotropic effects. This clarification has been added to the exclusion criteria (Comment 3; page 3, lines 103–104).

Specific Comments by Section

Abstract

Comments 1: Lines 21-22, Page 1: The assertion that "pain intensity significantly predicted both psychopathological symptomatology and sleep quality" overstates the conclusions that can be drawn from cross-sectional data. Such a design cannot establish predictive relationships or causality.

Authors' response: We thank the reviewer for this valuable observation. We agree that predictive or causal claims are not appropriate in the context of a cross-sectional design. Accordingly, we have revised the statement to reflect the nature of our analyses more accurately. These changes can be found on page 1, lines 24–26.

Original text: Regression analyses revealed that pain intensity significantly predicted both psychopathological symptomatology and sleep quality after controlling for sex, age, and pain duration.

Modified text: Hierarchical regression analyses revealed that pain intensity was a significant explanatory variable for both psychological distress and sleep quality (p < 0.05 across all models), even after statistically controlling for sex, age, and pain duration.

Furthermore, we have applied these terminological adjustments consistently across the Abstract, Results, Discussion, and Conclusion sections to avoid any misleading implications of causality. Specifically, we replaced terms such as “predicts” with more appropriate expressions like “explains” or “accounted for”, in line with the limitations of a cross-sectional design.

Introduction

Comments 2: Lines 82-84, Page 2: The justification for the age range, citing "epidemiological evidence indicating that chronic musculoskeletal pain is particularly prevalent in older adults," doesn't adequately support the inclusion of young adults (18 years) in the study.

Authors' response: We have clarified the justification for the selected age range. Although chronic musculoskeletal pain is more prevalent in older adults, several studies have also reported its presence in younger individuals, including those in early adulthood (de Souza et al., 2019). Including participants from 18 to 80 years allowed us to capture the full clinical spectrum of this condition as it presents across adulthood. Nevertheless, the actual age range in our sample was 41 to 75 years, which closely reflects the typical epidemiological profile of chronic musculoskeletal pain. Additionally, no significant age differences were observed between the CPMP and control groups. These changes can be found on pages 2-3, lines 94–97.

Original text: This age range was selected based on epidemiological evidence indicating that chronic musculoskeletal pain is particularly prevalent in older adults, with a significant increase in pain from mid-adulthood to the age of 80.

Modified text: This age range was selected based on epidemiological evidence indicating that chronic musculoskeletal pain can occur in younger adults and becomes increasingly prevalent and persistent with age, particularly from mid-adulthood to the age of 80.

References supporting the authors' response:

de Souza, I. M. B., Sakaguchi, T. F., Yuan, S. L. K., Matsutani, L. A., do Espírito-Santo, A. S., Pereira, C. A. B., & Marques, A. P. (2019). Prevalence of low back pain in the elderly population: a systematic review. Clinics (Sao Paulo, Brazil)74, e789. https://doi.org/10.6061/clinics/2019/e789

Methods - Participants

Comments 3: Lines 88-92, Page 2: The exclusion criterion for "psychotropic or neurological medications" appears inconsistently applied, especially considering that 86.7% of participants were on pain medications. Many of these medications possess psychoactive properties, which could easily confound psychological assessments.

Authors' response: We have clarified the exclusion criterion to avoid any confusion. Psychotropic or neurological medications were considered those with primary psychoactive effects (e.g., antidepressants, antipsychotics, anticonvulsants, or benzodiazepines), as these could influence the psychological variables assessed. The medications taken by participants in this study—mainly paracetamol and non-steroidal anti-inflammatory drugs (NSAIDs) such as ibuprofen, enantyum, aceclofenac, and diclofenac—are standard analgesics without clinically relevant psychotropic effects (Enthoven et al., 2016; Machado et al., 2015). Therefore, their use did not conflict with the exclusion criteria.

We have accordingly clarified in the exclusion criteria which types of medications were excluded. These changes can be found on page 3, lines 103–104.

Original text: Patients with CPMP were excluded if they (1) had been experiencing pain for less than three months, (2) had a medical or psychiatric condition, injury, or accident that could better explain their symptoms, (3) were taking psychotropic or neurological medications, whether for the treatment of chronic pain or for other unrelated conditions, or (4) had received any physical or psychological intervention prior to the study.

Modified text: Patients with CPMP were excluded if they (1) had been experiencing pain for less than three months, (2) had a medical or psychiatric condition, injury, or accident that could better explain their symptoms, (3) were taking psychotropic or neurological medications (e.g., antidepressants, antipsychotics, anticonvulsants, or benzodiazepines), whether for the treatment of chronic pain or for other unrelated conditions, or (4) had received any physical or psychological intervention prior to the study.

References supporting the authors' response:

Enthoven, W. T., Roelofs, P. D., Deyo, R. A., van Tulder, M. W., & Koes, B. W. (2016). Non-steroidal anti-inflammatory drugs for chronic low back pain. The Cochrane database of systematic reviews2(2), CD012087. https://doi.org/10.1002/14651858.CD012087

Machado, G. C., Maher, C. G., Ferreira, P. H., Pinheiro, M. B., Lin, C. W., Day, R. O., McLachlan, A. J., & Ferreira, M. L. (2015). Efficacy and safety of paracetamol for spinal pain and osteoarthritis: systematic review and meta-analysis of randomised placebo controlled trials. BMJ (Clinical research ed.)350, h1225. https://doi.org/10.1136/bmj.h1225

Comments 4: Lines 94-98, Page 2: The recruitment method introduces systematic bias by selecting CPMP patients from a specialized clinical setting while controls were recruited through community advertisements. This creates fundamental differences in healthcare-seeking behavior and baseline symptom severity between the groups.

Authors' response: We thank the reviewer for this thoughtful observation. We acknowledge that recruiting patients from a specialized clinical setting while recruiting controls from the community may introduce differences in healthcare-seeking behavior. However, this approach was necessary to ensure that all individuals in the CPMP group met ICD-11 diagnostic criteria, which require clinical evaluation by healthcare professionals. Since the collaborating center only provides care to clinical populations, it was not feasible to recruit healthy controls from the same setting. Therefore, controls were recruited through community advertisements and were screened to confirm the absence of chronic pain or psychiatric conditions.

The primary focus of the study was to characterize the psychological and sleep-related profile of individuals with CPMP. The healthy control group served as a baseline comparison to contextualize the findings in the CPMP group, rather than as a clinically matched sample. Although the recruitment sources differed, the two groups were comparable in terms of age, civil status, and employment situation, with no statistically significant differences in these variables. Differences in sex distribution were noted and were statistically controlled for in the analyses. We have added this limitation on page 9, lines 343–345:

Added text: A further consideration is the use of different recruitment sources for each group, which may have introduced systematic bias despite screening procedures and the statistical adjustments made to allow for between-group comparisons.

Methods - Statistical Analysis

Comments 5: Lines 158-159, Page 4: The Benjamini-Hochberg correction was applied selectively to group comparisons but not to correlation or regression analyses. This inconsistency undermines robust protection against Type I error across all statistical tests performed.

Authors' response: As recommended, we have applied the Benjamini-Hochberg correction to all statistical analyses, including group comparisons, correlation, and regression analyses, to ensure consistent control of Type I error. We have accordingly revised the tables and the Results section to reflect the adjusted p-values. In addition, the Data Analysis section has been updated to explicitly state the use of the Benjamini-Hochberg procedure across all statistical tests. These changes can be found in the Methods section, page 4, lines 181–183, and in the Results section.

Original text: To control for multiple comparisons and reduce the risk of Type I errors, the Benjamini-Hochberg correction (FDR = 0.05) was applied to adjust the p-values.

Modified text: To control for multiple comparisons and reduce the risk of Type I errors, the Benjamini-Hochberg correction (FDR = 0.05) was applied to all statistical analyses.

Comments 6: Lines 170-178, Page 4: Conducting a post-hoc power analysis instead of an a priori sample size calculation reflects poor study design. The reported effect sizes might be inflated due to the small sample size and existing selection bias.

Authors' response: We acknowledge the reviewer’s concern and have clarified that the power analysis was conducted post hoc. Due to the characteristics of the sample and the practical challenges in recruiting patients who met the diagnostic criteria for CPMP, we relied on sample sizes used in previous studies as a reference [32–34]. Furthermore, we have revised the Limitations section to explicitly recognize that the absence of an a priori power calculation is a limitation and that the reported effect sizes may be inflated due to the small sample size and potential selection bias. These changes can be found on page 9, lines 334–338.

Original text: Although the post hoc power analysis indicated sufficient power to detect the observed effects, future studies with larger samples would strengthen the robustness and replicability of these findings.

Modified text: Although the post hoc power analysis indicated sufficient power to detect the observed effects, the absence of an a priori power calculation limits the ability to draw definitive conclusions regarding sample adequacy. Future studies with larger samples would strengthen the robustness and replicability of these findings.

Results

Comments 7: Lines 184-187, Page 4: The significant gender difference between groups (p=0.019) is a critical confounding variable. This directly undermines the validity of all subsequent comparisons, given the well-established and substantial gender differences in psychological distress.

Authors' response: We acknowledge that the sex imbalance between the CPMP and control groups represents a potential confounding variable. To mitigate its impact, we statistically controlled for sex in all correlation and regression analyses. Additionally, we have now addressed this issue more explicitly in the Limitations section, acknowledging that statistical control may not fully account for differences in group composition. These clarifications have been added to page 9, lines 338–342.

Original text: Another limitation concerns the gender imbalance between groups, as the CPMP group included more women than the control group. Since women typically report higher levels of psychological distress, this may have influenced the results. However, gender was included as a covariate in the analyses to account for this potential effect.

Modified text: Another limitation concerns the sex imbalance between groups, as the CPMP group included more women than the control group. Since women typically report higher levels of psychological distress, this may have influenced the results. Although sex was included as a covariate in the analyses to account for this potential effect, this statistical control may not fully eliminate the influence of group composition on the findings.

Comments 8: Lines 188-193, Page 4: The description of pain characteristics is missing crucial clinical details such as the distribution of pain duration, comorbid conditions, and specific medication dosages. These details are vital, as they could influence reported psychological and sleep outcomes.

Authors' response: We appreciate the reviewer’s observation. As reported in the manuscript (page 5, lines 203-205), the distribution of pain duration in the CPMP group was described using the mean (11.95 years), standard deviation (SD = 1.85), and range (1–40 years), providing a clear overview of chronicity. Regarding comorbid conditions, all participants were screened for exclusion criteria, which explicitly ruled out individuals with any medical or psychiatric condition that could better explain their symptoms. Lastly, although information on medication dosages was not collected, participants taking psychotropic or neurological medications were excluded. Only non-psychotropic pain medications (e.g., paracetamol, ibuprofen) were listed in Table 1.

Comments 9: Lines 202-204, Page 5 (Statistical Results): The claim that "all dimensions" showed significant differences is inaccurate. Specifically, paranoid ideation showed no significant difference (p=0.197), an exception that is understated by being buried in the text rather than prominently discussed.

Authors' response: As suggested, we have revised the text to clearly indicate that paranoid ideation was the only SCL-90-R dimension without a statistically significant group difference, ensuring that this exception is explicitly stated rather than embedded in the narrative. In addition, following a separate reviewer’s recommendation, we have removed specific statistical values from the Results section, as these are already presented in the corresponding tables. These changes have been incorporated on page 6, lines 217–220.

Original text: SCL-90-R scores revealed statistically significant differences across all dimensions (all p < 0.05), except for paranoid ideation (p = 0.197), with the CPMP group scoring significantly higher on eight dimensions and all three global indices of distress.

Modified text: The only SCL-90-R dimension that did not show a statistically significant difference between groups was paranoid ideation. All other dimensions and global indices revealed statistically significant differences, with the CPMP group scoring higher on all remaining scales.

Comments 10: Lines 247-260, Page 7: The hierarchical regression analysis purports to control for confounding variables but notably fails to adequately address the fundamental gender imbalance. This imbalance likely accounts for a significant portion of the observed variance in psychological outcomes.

Authors' response: We acknowledge the reviewer’s concern regarding the potential impact of sex imbalance on psychological outcomes. To address this, sex was included as a covariate in the hierarchical regression analyses to statistically control for its influence, along with age and pain duration. While this approach helps mitigate the confounding effect, we recognize that statistical control may not fully account for the variance explained by differences in group composition. This limitation has now been explicitly acknowledged in the revised Limitations section. These changes can be found on page 9, lines 338–342.

Original text: Another limitation concerns the gender imbalance between groups, as the CPMP group included more women than the control group. Since women typically report higher levels of psychological distress, this may have influenced the results. However, gender was included as a covariate in the analyses to account for this potential effect.

Modified text: Another limitation concerns the sex imbalance between groups, as the CPMP group included more women than the control group. Since women typically report higher levels of psychological distress, this may have influenced the results. Although sex was included as a covariate in the analyses to account for this potential effect, this statistical control may not fully eliminate the influence of group composition on the findings.

Discussion

Comments 11: Lines 295-298, Page 8: The assertion that findings were obtained "after controlling for sex, age, and pain duration" is misleading. While statistical control was attempted, it cannot fully account for systematic group differences in gender distribution stemming from the initial recruitment.

Authors' response: As suggested, we have revised the phrasing to clarify that statistical control was applied. The text has been modified to state that analyses were conducted after statistically controlling for sex, age, and pain duration. These changes can be found on page 8, lines 301–303.

Original text: Regarding the relationship between pain intensity, psychological distress, and sleep quality in the CPMP group, our results showed that, after controlling for sex, age, and pain duration…

Modified text: Regarding the relationship between pain intensity, psychological distress, and sleep quality in the CPMP group, our results showed that, after statistically controlling for sex, age, and pain duration…

Additionally, the Abstract and Discussion sections have been revised to ensure consistency and to explicitly state that statistical control was applied.

Comments 12: Lines 318-332, Page 8: The acknowledgment of limitations related to the cross-sectional design directly contradicts earlier claims about "predictive" relationships made throughout the manuscript.

Authors' response: We agree that claims about prediction or causality are not appropriate given the cross-sectional nature of our study. Accordingly, we have carefully revised the manuscript to ensure consistency in the terminology used throughout. Specifically, we have replaced terms such as “predicts” with more appropriate expressions, such as “explains” or “accounted for”. These adjustments have been applied consistently across the Abstract, Results, Discussion, and Conclusion sections to avoid any misleading implications regarding causality.

Comments 13: Lines 325-328, Page 8: The discussion of gender imbalance as a limitation understates its profound impact by suggesting that simple statistical control can adequately resolve such a fundamental design flaw.

Authors' response: We appreciate the reviewer’s observation and understand that the sex imbalance may have had a substantial impact on the psychological outcomes observed. This issue has already been addressed in our responses to previous comments (Comments 7 and 10), and the manuscript has been revised accordingly. Specifically, we have clarified in the Limitations section that statistical control for sex may not fully account for the group imbalance, acknowledging the potential impact of this design limitation. These changes can be found on page 9, lines 338–342.

Original text: Another limitation concerns the gender imbalance between groups, as the CPMP group included more women than the control group. Since women typically report higher levels of psychological distress, this may have influenced the results. However, gender was included as a covariate in the analyses to account for this potential effect.

Modified text: Another limitation concerns the sex imbalance between groups, as the CPMP group included more women than the control group. Since women typically report higher levels of psychological distress, this may have influenced the results. Although sex was included as a covariate in the analyses to account for this potential effect, this statistical control may not fully eliminate the influence of group composition on the findings.

We would also like to highlight that the recruitment of participants meeting both the ICD-11 criteria for CPMP and our specific inclusion criteria presented practical challenges, due to the novelty of the diagnosis and the requirement of clinical confirmation. Consequently, a non-probabilistic sampling method was employed, which may have contributed to the observed sex imbalance and limited the representativeness of the sample. This contextual factor has also been discussed in the revised manuscript on page 9, lines 329–332.

Original text: Additionally, given the characteristics of the target population, a non-probabilistic sampling method was employed, which may have limited the representativeness of the sample.

Modified text: Additionally, given the characteristics of the target population and the practical challenges of recruiting participants who met the ICD-11 criteria for CPMP, a non-probabilistic sampling method was employed, which may have limited the representativeness of the sample.

Limitations

Comments 14: Lines 329-332, Page 8: Characterizing self-report bias as a minor limitation significantly understates the threat to validity when all primary outcomes are based exclusively on subjective measures without objective corroboration.

Authors' response: We have expanded the limitations associated with relying exclusively on self-report measures, emphasizing their subjective nature and the potential impact on data validity. In response to this comment, we now suggest the inclusion of objective assessment tools, such as actigraphy or clinician-rated measures, in future studies. This revision can be found on page 9, lines 345–351.

Original text: Finally, the assessment of psychopathological symptomatology, sleep quality, and pain intensity relied on self-report measures, which may be susceptible to response biases, such as the tendency to answer in a socially desirable manner.

Modified text: Finally, the assessment of psychopathological symptomatology, sleep quality, and pain intensity relied on self-report measures, which may be susceptible to response biases and reflect subjective experiences influenced by individual interpretation or emotional state, potentially compromising data reliability. Including objective methods in future studies, such as actigraphy for sleep or clinician-rated assessments for psychopathology and pain, could help improve the validity of the findings [52].

Added reference:

  1. Smith, M.T.; McCrae, C.S.; Cheung, J.; Martin, J.L.; Harrod, C.G.; Heald, J.L.; Carden, K.A. Use of Actigraphy for the Evaluation of Sleep Disorders and Circadian Rhythm Sleep-Wake Disorders: An American Academy of Sleep Medicine Systematic Review, Meta-Analysis, and GRADE Assessment. J Clin Sleep Med 2018, 14, 1209–1230, doi:10.5664/jcsm.7228.

Comments on the Quality of English Language: The manuscript is written with adequate scientific clarity and generally uses appropriate terminology. However, it frequently overstates the strength of evidence that can legitimately be drawn from its cross-sectional design.

Authors' response: We thank the reviewer for this observation. In response, we have thoroughly revised the manuscript to ensure that all interpretations are consistent with the limitations inherent to a cross-sectional design. In particular, we have tempered any overstatements by replacing terms that could imply prediction or causality with more appropriate expressions, such as “explains” or “is associated with”. These adjustments have been consistently implemented across the Abstract, Results, Discussion, and Conclusion sections to avoid overstating the strength of the evidence.

Reviewer 3 Report

Comments and Suggestions for Authors

Introduction:

Justify the prevalence of sleep disturbance in the process of chronic musculoskeletal pain.

Materials and Methods

Very wide age range. The authors do not record employment activity. This is a key element in this type of process. Justify the minimum age range or modify it based on your argumentation on line 84.

Line 119 y 120: It would be necessary to identify what these questionnaires measure as has been done for the other variables.

From a biopsychosocial perspective, the assessment of health-related quality of life is lacking, a variable in line with the proposed instruments and indicated in chronic musculoskeletal pain processes. Please justify its absence.

Results:

Due to the wide age range, it would be interesting to know the results broken down by age group.

It's striking that the study subjects with chronic musculoskeletal pain were not prescribed medication for psychological distress. Can the authors justify this?

The sample of subjects is 100% chronic low back pain. This aspect is relevant to the focus of the title and introduction of the manuscript. The authors should reconsider this.

Author Response

REVIEWER 3#

Introduction

Comments 1: Justify the prevalence of sleep disturbance in the process of chronic musculoskeletal pain.

Authors' response: We thank the reviewer for this suggestion. In response, we have revised the introduction to better justify the prevalence and clinical relevance of sleep disturbances in the context of chronic musculoskeletal pain. Specifically, we now highlight that sleep disturbances are among the most prevalent and disabling comorbid symptoms in this population and are increasingly recognized as a core component of the psychological burden associated with chronic pain. We have also incorporated recent evidence supporting a bidirectional relationship between sleep quality and pain-related psychological distress, reinforcing the importance of addressing sleep in this clinical context. These changes can be found in the revised paragraph on page 2, lines 58–63.

Original text: Recent systematic reviews have reported elevated rates of psychopathological symptoms among individuals with CMP, including depression, anxiety, distress, somatization, and sleep disturbances. These symptoms, when persistent over time, may increase the risk of pain chronification [11,13,14]. Nevertheless, the relationship between pain intensity and psychological symptomatology in CMP remains unclear. While some studies have found that psychological symptoms predict pain intensity, less is known about whether pain intensity itself can predict psychopathological symptoms [7,8,11,12].

Modified text: Recent systematic reviews have reported elevated rates of psychopathological symptoms among individuals with CMP, including depression, anxiety, distress, somatization, and sleep disturbances. These symptoms, when persistent over time, may increase the risk of pain chronification [11,13,14]. Among them, sleep disturbances are especially prevalent and disabling, and are increasingly recognized as a core component of the psychological burden associated with chronic pain [15,16]. Far from being a mere consequence of pain, recent studies suggest that poor sleep quality may intensify psychological distress by impairing emotional regulation and cognitive functioning, thereby reinforcing the cycle of pain and emotional suffering [17,18]. Nevertheless, the relationship between pain intensity and psychological symptomatology in CMP remains unclear. While some studies have found that psychological symptoms predict pain intensity, less is known about whether pain intensity itself can explain psychopathological symptoms [7,8,11,12].

Added references:

  1. Goossens, Z.; Van Stallen, A.; Vermuyten, J.; De deyne, M.; Rice, D.; Runge, N.; Huysmans, E.; Vantilborgh, T.; Nijs, J.; Mairesse, O.; et al. Day-to-Day Associations between Pain Intensity and Sleep Outcomes in an Adult Chronic Musculoskeletal Pain Population: A Systematic Review. Sleep Medicine Reviews 2025, 79, 102013, doi:10.1016/j.smrv.2024.102013.
  2. Jain, S.V.; Panjeton, G.D.; Martins, Y.C. Relationship Between Sleep Disturbances and Chronic Pain: A Narrative Review. Clinics and Practice 2024, 14, 2650–2660, doi:10.3390/clinpract14060209.
  3. Finan, P.H.; Goodin, B.R.; Smith, M.T. The Association of Sleep and Pain: An Update and a Path Forward. J Pain 2013, 14, 1539–1552, doi:10.1016/j.jpain.2013.08.007.
  4. Seiger, A.N.; Penzel, T.; Fietze, I. Chronic Pain Management and Sleep Disorders. Cell Reports Medicine 2024, 5, 101761, doi:10.1016/j.xcrm.2024.101761.

Materials and Methods

Comments 2: Very wide age range. The authors do not record employment activity. This is a key element in this type of process. Justify the minimum age range or modify it based on your argumentation on line 84.

Authors' response: We have clarified the justification for the selected age range in the revised manuscript. Although chronic musculoskeletal pain is more prevalent in older adults, studies have also reported its presence in younger individuals, including those in early adulthood (de Souza et al., 2019). Including participants from 18 to 80 years allowed us to capture this broader clinical spectrum. Nevertheless, the actual age range in our sample was from 41 to 75 years, which closely aligns with the typical epidemiological profile of chronic musculoskeletal pain. Additionally, no significant differences were observed between the CPMP and control groups in terms of age. This clarification has been added on page 2-3, lines 94–97.

Original text: This age range was selected based on epidemiological evidence indicating that chronic musculoskeletal pain is particularly prevalent in older adults, with a significant increase in pain from mid-adulthood to the age of 80.

Modified text: This age range was selected based on epidemiological evidence indicating that chronic musculoskeletal pain can occur in younger adults and becomes increasingly prevalent and persistent with age, particularly from mid-adulthood to the age of 80.

Furthermore, in response to the reviewer’s observation regarding employment activity, we have now included the variable employment situation in the results table and added the following clarification in the text on page 5, lines 201–202:

Original text: No significant differences were observed in marital status.

Modified text: No significant differences were observed in marital status or employment situation.

References supporting the authors' response:

de Souza, I. M. B., Sakaguchi, T. F., Yuan, S. L. K., Matsutani, L. A., do Espírito-Santo, A. S., Pereira, C. A. B., & Marques, A. P. (2019). Prevalence of low back pain in the elderly population: a systematic review. Clinics (Sao Paulo, Brazil)74, e789. https://doi.org/10.6061/clinics/2019/e789

Comments 3: Line 119 y 120: It would be necessary to identify what these questionnaires measure as has been done for the other variables.

Authors' response: We have revised the manuscript to clarify what each of the three global indices of the SCL-90-R measures. This specification has been added on page 3, lines 131–135.

Original text: Additionally, the scale includes three global distress indices: the Global Severity Index (GSI), the Positive Symptom Total (PST), and the Positive Symptom Distress Index (PSDI).

Modified text: Additionally, the scale includes three global distress indices: the Global Severity Index (GSI), which reflects overall psychological distress; the Positive Symptom Total (PST), which indicates the number of reported symptoms regardless of intensity; and the Positive Symptom Distress Index (PSDI), which assesses the average intensity of the reported symptoms.

Comments 4: From a biopsychosocial perspective, the assessment of health-related quality of life is lacking, a variable in line with the proposed instruments and indicated in chronic musculoskeletal pain processes. Please justify its absence.

Authors response: We thank the reviewer for this relevant observation. We acknowledge the importance of assessing health-related quality of life (HRQoL) within the biopsychosocial framework, particularly in chronic musculoskeletal pain research. However, as chronic primary musculoskeletal pain (CPMP) is a relatively new diagnostic category introduced in the ICD-11, this study was conceived as an exploratory investigation focused on the psychological and sleep-related correlates of pain. Given the extensive number of instruments included and the exploratory nature of the study, we chose to prioritize core psychopathological and sleep variables. Nevertheless, we agree that the inclusion of HRQoL measures would be valuable in providing a more comprehensive understanding of the impact of CPMP. We have therefore included this recommendation in the discussion as a future line of research, on pages 9-10, lines 362-365.

Added text: In addition, future research should consider including measures of health-related quality of life, a relevant construct within the biopsychosocial model, which may complement the psychological and sleep-related variables examined in the present study.

Results:

Comments 5: Due to the wide age range, it would be interesting to know the results broken down by age group.

Authors' response: We thank the reviewer for this suggestion. Although the theoretical age range considered for inclusion was 18 to 80 years, the actual participants in the study were aged between 41 and 75 years. Moreover, no significant age differences were observed between the CPMP and control groups. Given the small sample size of the clinical group (n = 30), subdividing the data by age groups would have considerably reduced statistical power, limited the robustness of the results, and precluded any meaningful generalisation. Furthermore, age was not a primary variable of interest in this study, which was conceived as an exploratory investigation due to the recent introduction of the CPMP diagnosis in the ICD-11. The aim was to examine the psychological and sleep-related correlates of this condition. Nevertheless, to control for potential confounding effects, age was included as a covariate in all correlation and regression analyses, along with sex and pain duration. To address this issue, a clarification has been added to the limitations section on page 9, lines 332–334.

Original text: The relatively small sample size is another limitation, potentially affecting the generalizability of the results.

Modified text: The relatively small sample size is another limitation, potentially affecting the generalizability of the results and limiting the possibility of conducting subgroup analyses, such as comparisons across age groups.

Comments 6: It's striking that the study subjects with chronic musculoskeletal pain were not prescribed medication for psychological distress. Can the authors justify this?

Authors' response: Prior to the final selection of participants in the CPMP group, there were individuals who reported the use of medication for psychological distress. However, following the application of the exclusion criteria, we retained only those participants who were not taking any psychotropic or neurological medications, as these could have interfered with the assessment of psychological variables. We have clarified in the revised manuscript that psychotropic or neurological medications were defined as those with primary psychoactive effects (e.g., antidepressants, antipsychotics, anticonvulsants, or benzodiazepines). In contrast, the medications used by participants included only standard analgesics—such as paracetamol and non-steroidal anti-inflammatory drugs (NSAIDs) like ibuprofen, enantyum, aceclofenac, and diclofenac—which do not have clinically relevant psychotropic effects (Enthoven et al., 2016; Machado et al., 2015). Therefore, their use did not conflict with the study’s exclusion criteria. To avoid confusion, this clarification has been incorporated into the exclusion criteria on page 3, lines 103–104.

Original text: Patients with CPMP were excluded if they (1) had been experiencing pain for less than three months, (2) had a medical or psychiatric condition, injury, or accident that could better explain their symptoms, (3) were taking psychotropic or neurological medications, whether for the treatment of chronic pain or for other unrelated conditions, or (4) had received any physical or psychological intervention prior to the study.

Modified text: Patients with CPMP were excluded if they (1) had been experiencing pain for less than three months, (2) had a medical or psychiatric condition, injury, or accident that could better explain their symptoms, (3) were taking psychotropic or neurological medications (e.g., antidepressants, antipsychotics, anticonvulsants, or benzodiazepines), whether for the treatment of chronic pain or for other unrelated conditions, or (4) had received any physical or psychological intervention prior to the study

References supporting the authors' response:

Enthoven, W. T., Roelofs, P. D., Deyo, R. A., van Tulder, M. W., & Koes, B. W. (2016). Non-steroidal anti-inflammatory drugs for chronic low back pain. The Cochrane database of systematic reviews2(2), CD012087. https://doi.org/10.1002/14651858.CD012087

Machado, G. C., Maher, C. G., Ferreira, P. H., Pinheiro, M. B., Lin, C. W., Day, R. O., McLachlan, A. J., & Ferreira, M. L. (2015). Efficacy and safety of paracetamol for spinal pain and osteoarthritis: systematic review and meta-analysis of randomised placebo controlled trials. BMJ (Clinical research ed.)350, h1225. https://doi.org/10.1136/bmj.h1225

Comments 7: The sample of subjects is 100% chronic low back pain. This aspect is relevant to the focus of the title and introduction of the manuscript. The authors should reconsider this.

Authors' response: We thank the reviewer for this thoughtful comment. While it is true that all participants in the CPMP group reported chronic low back pain, the study was framed within the broader diagnostic category of Chronic Primary Musculoskeletal Pain (CPMP), as defined by the ICD-11. This decision was based on the exploratory nature of the study and the current lack of research specifically focused on CPMP as a diagnostic entity. Chronic primary low back pain represents one of the most common manifestations within this category; however, given the novelty of this classification, we sought to examine the broader construct rather than limit the investigation to a single anatomical location.

The inclusion criteria, recruitment procedures, and clinical confirmation process were specifically designed to align with the diagnostic requirements for CPMP. Although the final sample consisted exclusively of individuals with low back pain, the aim was to explore psychological and sleep-related features associated with CPMP more generally. Accordingly, both the title and the abstract have been revised to reflect this broader conceptual framework, in line with the ICD-11 definition.

Original title: Psychopathological symptomatology and sleep quality in chronic pain: a comparison with healthy controls

Modified title: Psychopathological symptomatology and sleep quality in chronic primary musculoskeletal pain: a comparison with healthy controls

Round 2

Reviewer 1 Report

Comments and Suggestions for Authors

Thank you for the careful and informative revisions you made, taking into account the reviewer's comments.

I believe the changes made have positively contributed to the quality of the study.

I wish you continued success.

Author Response

The authors are grateful for the comments of reviewer 1, which have undoubtedly contributed to the improvement of this article.

Reviewer 2 Report

Comments and Suggestions for Authors

GENERAL COMMENTS

I appreciate the improvements the authors have made to the manuscript since the last review. The study explores an important clinical question regarding the psychological and sleep-related aspects of chronic primary musculoskeletal pain (CPMP), utilizing the ICD-11 classification.

While the overall methodology is sound , there are several methodological and presentation points that still need attention before publication.

SPECIFIC COMMENTS

Abstract:

Page 1, Line 13: The phrase "of unknown aetiology" is redundant here, as CPMP is defined by ICD-11 criteria as having no clearly identifiable medical cause.

Page 1, Line 14: The statement "limited research on chronic primary musculoskeletal pain (CPMP)" requires supporting evidence. Alternatively, it could be qualified as "relatively limited" with appropriate justification.

Page 1, Line 24: The correlation coefficients should be reported consistently with the same number of decimal places (e.g., "r = 0.785" vs "r = 0.83").

Introduction:

Page 2, Lines 58-66: The highlighted text in yellow indicates this might be a draft version and should be cleaned up for submission.

Page 2, Line 94: The justification "This age range was selected based on epidemiological evidence" seems weak. More specific rationale for the upper age limit of 80 years should be provided.

Methods:

Page 3, Lines 103-104: The exclusion criterion regarding "psychotropic or neurological medications" is quite broad. This might limit the representativeness of the chronic pain population, where such medications are commonly prescribed.

Page 4, Lines 185-192: While a post-hoc power analysis is mentioned, there's no indication of an a priori power calculation for sample size determination, which is a methodological weakness.

Page 4, Line 178: "explanatory variable" should be revised to "predictor variable," as the cross-sectional design cannot establish causation.

Results:

Page 5, Table 1: The table formatting could be improved by removing the repetition of "Employment situation, n (%):" and ensuring consistent spacing.

Page 6, Lines 216-233: This paragraph largely repeats information already presented in Table 2, leading to unnecessary redundancy.

Page 7, Line 260: Similar to the point above, "explanatory variable" should be changed to "predictor variable" given the study design.

Discussion:

Page 8, Lines 301-302: The phrase "statistically controlling for sex, age, and pain duration" is repetitive, as this was already established in the methods section.

Page 9, Lines 329-330: The statement about "practical challenges of recruiting participants who met the ICD-11 criteria" needs further elaboration. What specific challenges were encountered?

Page 9, Lines 332-333: When discussing the "relatively small sample size" limitation, it should acknowledge the potential impact on statistical power for any subgroup analyses that were not performed.

Page 10, Lines 363-365: The suggestion for "measures of health-related quality of life" is mentioned but not well-integrated into the broader discussion of how this would enhance the biopsychosocial understanding of CPMP.

Statistical Analysis:

Page 4, Lines 176-180: The description of the hierarchical regression analysis needs clarification regarding whether the covariates were entered in the first step and pain intensity in the second step.

Page 7, Table 4: The R2 values should be reported as adjusted R2, given that covariates were included in the model.

Comments on the Quality of English Language

Writing Quality:

Page 8, Line 275: "Given the growing body of evidence" — please specify a time frame or quantify this statement.

Page 9, Line 311: "significantly explained" should be "significantly predicted" due to the cross-sectional design.

Page 10, Line 364: "a relevant construct within the biopsychosocial model" — this phrase is vague and should be more specific about how quality of life measures would complement the current findings.

Author Response

Abstract

Comments 1: Page 1, Line 13: The phrase "of unknown aetiology" is redundant here, as CPMP is defined by ICD-11 criteria as having no clearly identifiable medical cause.

Authors' response: We thank the reviewer for this helpful observation. Accordingly, we have revised the sentence to reflect the ICD-11 definition of CPMP. These changes can be found on page 1, lines 13–14.

Original text: Chronic musculoskeletal pain of unknown aetiology is characterised by significant emotional distress and/or functional disability.

Modified text: Chronic musculoskeletal pain without a clearly identifiable medical cause is characterised by significant emotional distress and/or functional disability.

Comments 2: Page 1, Line 14: The statement "limited research on chronic primary musculoskeletal pain (CPMP)" requires supporting evidence. Alternatively, it could be qualified as "relatively limited" with appropriate justification.

Authors' response: We have revised the statement to clarify that the limited research refers specifically to studies on CPMP as defined in the latest revision of the ICD-11. This clarification reflects the relatively recent conceptualization of CPMP as a distinct diagnostic category and highlights the need for more focused investigations within this framework. These changes can be found on page 1, lines 14–18.

Original text: Given the limited research on chronic primary musculoskeletal pain (CPMP), the present study aimed to examine its psychopathological and sleep-related implications, and to explore whether pain intensity is associated with psychological distress and poor sleep quality.

Modified text: Given the relatively limited research specifically addressing chronic primary musculoskeletal pain (CPMP), as defined in the latest revision of the International Classification of Diseases (ICD-11), the present study aimed to examine its psychopathological and sleep-related implications, and to explore whether pain intensity is associated with psychological distress and poor sleep quality.

Comments 3: Page 1, Line 24: The correlation coefficients should be reported consistently with the same number of decimal places (e.g., "r = 0.785" vs "r = 0.83").

Authors' response: We have revised the text to report all correlation coefficients using three decimal places for consistency. Additionally, we have corrected the specific value mentioned; the correct coefficient is r = 0.838. This change can be found on page 1, lines 24-25.

Original text: Greater pain intensity on the NPRS was strongly associated with psychological distress (e.g., GSI: r = 0.83, p < 0.01) and poor sleep quality (r = 0.785, p < 0.01).

Modified text: Greater pain intensity on the NPRS was strongly associated with psychological distress (e.g., GSI: r = 0.838, p < 0.01) and poor sleep quality (r = 0.785, p < 0.01).

Introduction

Comments 4: Page 2, Lines 58-66: The highlighted text in yellow indicates this might be a draft version and should be cleaned up for submission.

Authors' response: We thank the reviewer for this observation. The yellow highlighting present in the previous version has been removed in the revised manuscript. Only the text newly added in response to the current round of revisions remains highlighted, in accordance with resubmission instructions.

Comments 5: Page 2, Line 94: The justification "This age range was selected based on epidemiological evidence" seems weak. More specific rationale for the upper age limit of 80 years should be provided.

Authors' response: We have revised the sentence to provide a more specific justification for the selected age range. The updated version clarifies the increasing prevalence and persistence of chronic musculoskeletal pain with advancing age and justifies the upper limit of 80 years by considering the higher likelihood of multimorbidity and disability in older populations (de Souza et al., 2019). This change can be found on page 3, lines 94–98.

Original text: This age range was selected based on epidemiological evidence indicating that chronic musculoskeletal pain can occur in younger adults and becomes increasingly prevalent and persistent with age, particularly from mid-adulthood to the age of 80.

Modified text: This age range was selected based on epidemiological evidence indicating that chronic musculoskeletal pain can occur in younger adults and becomes increasingly prevalent and persistent with age, particularly from mid-adulthood up to around the age of 80, before the prevalence of multimorbidity and functional disability becomes substantially higher.

References supporting the authors' response:

de Souza, I. M. B., Sakaguchi, T. F., Yuan, S. L. K., Matsutani, L. A., do Espírito-Santo, A. S., Pereira, C. A. B., & Marques, A. P. (2019). Prevalence of low back pain in the elderly population: a systematic review. Clinics (Sao Paulo, Brazil)74, e789. https://doi.org/10.6061/clinics/2019/e789

Methods

Comments 6: Page 3, Lines 103-104: The exclusion criterion regarding "psychotropic or neurological medications" is quite broad. This might limit the representativeness of the chronic pain population, where such medications are commonly prescribed.

Authors' response: We appreciate this observation. The exclusion of participants taking psychotropic or neurological medications was based on the specific aim of the study: to examine the psychopathological and sleep-related implications of CPMP. These types of medications can have relevant effects on psychological symptoms and sleep quality, potentially confounding the interpretation of our results. To preserve the internal validity of the study and to assess participants in their natural psychological and functional state, we opted for this criterion. We acknowledge that it introduced practical challenges during recruitment and may have limited the representativeness of the sample. These challenges are reflected in the general discussion of recruitment difficulties in the Limitations section. However, standard analgesics without clinically relevant psychotropic effects (e.g., paracetamol, NSAIDs) were permitted, in line with common clinical practice and to maintain feasibility in participant recruitment.

Comments 7: Page 4, Lines 185-192: While a post-hoc power analysis is mentioned, there's no indication of an a priori power calculation for sample size determination, which is a methodological weakness.

Authors' response: Although a post hoc power analysis was conducted and indicated sufficient power to detect the observed effects, we acknowledge that the absence of an a priori power calculation represents a methodological limitation. Accordingly, we have revised the text to explicitly state that no a priori power analysis was conducted and to clarify that the sample size was based on previous studies with comparable designs. This change can be found on page 4, lines 191–194.

Original text: To evaluate the statistical power of our analyses, we conducted a post hoc power analysis. With a significance level of α = 0.05, we compared psychological distress (SCL-90-R GSI) between chronic pain patients and control participants, obtaining a large effect size (d = 0.74) with high statistical power (1 − β = 0.78). Similarly, for sleep quality (PSQI), the effect size was also large (d = 0.61), with moderate power (1 − β = 0.62). These results indicate that the study had sufficient power to detect the observed effects. The sample size was based on previous studies examining similar relationships between chronic pain and psychopathological symptoms, which used comparable sample sizes [32–34].

Modified text: To evaluate the statistical power of our analyses, we conducted a post hoc power analysis. With a significance level of α = 0.05, we compared psychological distress (SCL-90-R GSI) between chronic pain patients and control participants, obtaining a large effect size (d = 0.74) with high statistical power (1 − β = 0.78). Similarly, for sleep quality (PSQI), the effect size was also large (d = 0.61), with moderate power (1 − β = 0.62). These results indicate that the study had sufficient power to detect the observed effects. Although a power calculation was not conducted a priori, the sample size was based on previous studies examining similar relationships between chronic pain and psychopathological symptoms, which used comparable sample sizes [32–34].

Additionally, we have expanded the Limitations section to specify the methodological issues related to the lack of an a priori power analysis and to provide guidance for future studies. This change can be found on page 9, lines 334–339.

Original text: Although the post hoc power analysis indicated sufficient power to detect the observed effects, the absence of an a priori power calculation limits the ability to draw definitive conclusions regarding sample adequacy. Future studies with larger samples would strengthen the robustness and replicability of these findings.

Modified text: Although the post hoc power analysis indicated sufficient power to detect the observed effects, the lack of an a priori power calculation represents a methodological limitation and constrains the ability to draw definitive conclusions regarding sample adequacy. Future studies with larger samples and predefined sample size estimations would strengthen the robustness and replicability of these findings.

Comments 8: Page 4, Line 178: "explanatory variable" should be revised to "predictor variable," as the cross-sectional design cannot establish causation.

Authors' response: We have revised the term "explanatory variable" to "predictor variable" to reflect the cross-sectional nature of the study and to avoid causal implications. This change can be found on page 4, lines 178-180.

Original text: Finally, a hierarchical linear regression was conducted within the CPMP group, using pain intensity as the explanatory variable while controlling for sex, age, and pain duration…

Modified text: Finally, a hierarchical linear regression was conducted within the CPMP group to examine whether pain intensity remained a significant predictor of psychopathological symptomatology and sleep quality after controlling for sex, age, and pain duration.

Results

Comments 9: Page 5, Table 1: The table formatting could be improved by removing the repetition of "Employment situation, n (%):" and ensuring consistent spacing.

Authors' response: We have revised the table to remove "Employment situation, n (%)" and replaced it with the more suitable term "Occupational status, n (%)". We have also ensured consistent spacing and formatting throughout Table 1 to improve clarity and readability.

Comments 10: Page 6, Lines 216-233: This paragraph largely repeats information already presented in Table 2, leading to unnecessary redundancy.

Authors' response: We have streamlined the paragraph to reduce redundancy with Table 2 while preserving the key findings and maintaining the original phrasing as much as possible. The changes can be found on page 6, lines 218–230.

Original text: Results of the Mann-Whitney U test for SCL-90-R and PSQI scores between the chronic pain and control groups are presented in Table 2. The only SCL-90-R dimension that did not show a statistically significant difference between groups was paranoid ideation. All other dimensions and global indices revealed statistically significant differences, with the CPMP group scoring higher on all remaining scales. Within the CPMP group, scores for somatization were clinically significant, suggesting the presence of psychopathological symptoms. Scores for the obsessive-compulsive dimension approached clinical significance, while those for depression, interpersonal sensitivity, psychoticism, hostility, and anxiety fell within the normative range. Scores for paranoid ideation and phobic anxiety were below average. This group also obtained values close to clinical significance for the Global Severity Index (GSI) and the Positive Symptom Total (PST), and normative scores for the Positive Symptom Distress Index (PSDI). As expected, the HC group scored below the normative range across all dimensions and indices, indicating no signs of psychological distress.

Regarding the PSQI, significant group differences were observed in global scores, with the CPMP group reporting poorer sleep quality than the HC group. The CPMP group was classified as having moderately impaired sleep quality, whereas the HC group fell within the mildly impaired range.

Modified text: Results of the Mann-Whitney U test for SCL-90-R and PSQI scores between the chronic pain and control groups are presented in Table 2. The only SCL-90-R dimension that did not show a statistically significant difference between groups was paranoid ideation. All other dimensions and global indices revealed statistically significant differences, with the CPMP group scoring higher on all remaining scales. Within the CPMP group, somatization reached clinical significance, and obsessive-compulsive symptoms approached that threshold. The remaining dimensions fell within normative or below-average ranges. Scores on the GSI and PST were close to clinical significance, while PSDI scores were normative. The HC group scored below the normative range across all dimensions and indices, indicating no signs of psychological distress. Regarding the PSQI, significant group differences were observed in global scores, with the CPMP group reporting poorer sleep quality than the HC group. The CPMP group was classified as having moderately impaired sleep quality, whereas the HC group fell within the mildly impaired range.

Comments 11: Page 7, Line 260: Similar to the point above, "explanatory variable" should be changed to "predictor variable" given the study design.

Authors' response: We have replaced the term "explain" with "be a predictor variable" to reflect terminology more appropriate to the study design. The change can be found on page 7, lines 258-260.

Original text: Finally, to determine whether pain intensity could explain psychological distress and sleep quality in the CPMP group, a hierarchical linear regression analysis was conducted (Table 4).

Modified text: Finally, to determine whether pain intensity could be a predictor variable for psychological distress and sleep quality in the CPMP group, a hierarchical linear regression analysis was conducted (Table 4).

Discussion

Comments 12: Page 8, Lines 301-302: The phrase "statistically controlling for sex, age, and pain duration" is repetitive, as this was already established in the methods section.

Authors' response: We have removed the redundant reference to statistical control, in accordance with the reviewer’s suggestion. The changes can be found on page 8, lines 299–302.

Original text: Regarding the relationship between pain intensity, psychological distress, and sleep quality in the CPMP group, our results showed that, after statistically controlling for sex, age, and pain duration, higher pain intensity was significantly associated with greater psychopathological symptomatology across all SCL-90-R dimensions, as well as with poorer sleep quality.

Modified text: Regarding the relationship between pain intensity, psychological distress, and sleep quality in the CPMP group, our results showed that higher pain intensity was significantly associated with greater psychopathological symptomatology across all SCL-90-R dimensions, as well as with poorer sleep quality.

Comments 13: Page 9, Lines 329-330: The statement about "practical challenges of recruiting participants who met the ICD-11 criteria" needs further elaboration. What specific challenges were encountered?

Authors' response: We thank the reviewer for this comment. We have clarified the specific challenges involved in recruiting participants who met the ICD-11 diagnostic criteria for CPMP. The revised sentence specifies the difficulty in identifying cases of chronic pain not attributable to a known medical condition, as required by the ICD-11 definition. This change can be found on page 9, lines 327–331.

Original text: Additionally, given the characteristics of the target population and the practical challenges of recruiting participants who met the ICD-11 criteria for CPMP, a non-probabilistic sampling method was employed, which may have limited the representativeness of the sample.

Modified text: Additionally, given the characteristics of the target population and the difficulty in identifying cases of chronic musculoskeletal pain that met the ICD-11 criteria, which require the pain not be attributable to a known medical condition, a non-probabilistic sampling method was employed. This may have limited the representativeness of the sample, as all participants in the clinical group had low back pain.

Comments 14: Page 9, Lines 332-333: When discussing the "relatively small sample size" limitation, it should acknowledge the potential impact on statistical power for any subgroup analyses that were not performed.

Authors' response: We have explicitly acknowledged the potential impact of the small sample size on statistical power for subgroup analyses that were not performed, in accordance with the reviewer’s suggestion. The changes can be found on page 9, lines 331–334.

Original text: The relatively small sample size is another limitation, potentially affecting the generalizability of the results and limiting the possibility of conducting subgroup analyses, such as comparisons across age groups.

Modified text: The relatively small sample size is another limitation, potentially affecting the generalizability of the results and limiting the statistical power for conducting subgroup analyses, such as comparisons across age groups, which were not performed.

Comments 15: Page 10, Lines 363-365: The suggestion for "measures of health-related quality of life" is mentioned but not well-integrated into the broader discussion of how this would enhance the biopsychosocial understanding of CPMP.

Authors' response: This future research direction was suggested by a previous reviewer. We have now provided a clearer explanation of how the inclusion of health-related quality of life measures would enhance the biopsychosocial understanding of CPMP. The change can be found on pages 9-10, lines 363–368.

Original text: In addition, future research should consider including measures of health-related quality of life, a relevant construct within the biopsychosocial model, which may complement the psychological and sleep-related variables examined in the present study.

Modified text: In addition, future research should consider including measures of health-related quality of life, as these can capture the broader impact of chronic pain on daily functioning, social participation, and overall well-being, thereby complementing the psychological and sleep-related variables examined in the present study and contributing to a more comprehensive understanding of CPMP within the biopsychosocial model.

Statistical Analysis

Comments 16: Page 4, Lines 176-180: The description of the hierarchical regression analysis needs clarification regarding whether the covariates were entered in the first step and pain intensity in the second step.

Authors' response: We thank the reviewer for this observation. We have revised the sentence to clarify the order of entry in the hierarchical regression model. Specifically, the covariates (sex, age, and pain duration) were entered in the first step, followed by pain intensity in the second step. This change can be found on page 4, lines 178–182.

Original text: Finally, a hierarchical linear regression was conducted within the CPMP group, using pain intensity as the predictor variable while controlling for sex, age, and pain duration, to determine whether pain intensity remained a significant explanatory factor of psychopathological symptomatology and sleep quality after adjusting for these covariates.

Modified text: Finally, a hierarchical linear regression was conducted within the CPMP group to examine whether pain intensity remained a significant predictor of psychopathological symptomatology and sleep quality after controlling for sex, age, and pain duration. The covariates were entered in the first step of the model, and pain intensity was entered in the second step.

Comments 17: Page 7, Table 4: The R2 values should be reported as adjusted R2, given that covariates were included in the model.

Authors' response: We have revised Table 4 to report adjusted R² values, as suggested. The updated version can be found on page 7.

Writing Quality

Comments 18: Page 8, Line 275: "Given the growing body of evidence" — please specify a time frame or quantify this statement.

Authors' response: We have specified the time frame by referring to the growing body of evidence since the official adoption of the ICD-11 classification by the World Health Assembly in 2019 (World Health Organization, 2019). The change can be found on page 8, lines 274-277.

Original text: Given the growing body of evidence on chronic primary musculoskeletal pain (CPMP), especially within the framework of the ICD-11 classification, it becomes increasingly important to explore the psychological and functional consequences associated with this diagnosis.

Modified text: Given the growing body of evidence on chronic primary musculoskeletal pain (CPMP) since the official adoption of the ICD-11 classification in 2019, it becomes increasingly important to explore the psychological and functional consequences associated with this diagnosis [3,5].

References supporting the authors' response:

World Health Organization. (2019). The 72nd World Health Assembly resolution for ICD-11 adoption. https://www.who.int/publications/m/item/eleventh-revision-of-the-international-classification-of-diseases-adoption-wha72 (accessed July 22, 2025).

Comments 19: Page 9, Line 311: "significantly explained" should be "significantly predicted" due to the cross-sectional design.

Authors' response: We have replaced “significantly explained” with “significantly predicted”, in line with the reviewer’s suggestion. The change can be found on page 9, line 308.

Original text: Moreover, regression analyses confirmed that pain intensity significantly explained both psychological distress and sleep quality, even after adjusting for relevant covariates.

Modified text: Moreover, regression analyses confirmed that pain intensity significantly predicted both psychological distress and sleep quality, even after adjusting for relevant covariates.

Comments 20: Page 10, Line 364: "a relevant construct within the biopsychosocial model" — this phrase is vague and should be more specific about how quality of life measures would complement the current findings.

Authors' response: We have revised the sentence to clarify how measures of health-related quality of life would complement the current findings and contribute to a broader understanding of chronic pain in the context of the biopsychosocial model. The change can be found on pages 9-10, lines 363–368.

Original text: In addition, future research should consider including measures of health-related quality of life, a relevant construct within the biopsychosocial model, which may complement the psychological and sleep-related variables examined in the present study.

Modified text: In addition, future research should consider including measures of health-related quality of life, as these can capture the broader impact of chronic pain on daily functioning, social participation, and overall well-being, thereby complementing the psychological and sleep-related variables examined in the present study and contributing to a more comprehensive understanding of CPMP within the biopsychosocial model.

Reviewer 3 Report

Comments and Suggestions for Authors

I congratulate the authors for their work.
I thank the authors for their willingness to improve their work based on the reviewer's comments. Substantial and consistent changes can be observed in the work, with good justification from the authors.
Although the authors consider primary chronic musculoskeletal pain in their study, the overall sample refers to low back pain. Perhaps this aspect should be reflected at least in the study's limitations, since the term primary chronic musculoskeletal pain is a broader concept and encompasses other types of conditions.

Author Response

Comments 1: Although the authors consider primary chronic musculoskeletal pain in their study, the overall sample refers to low back pain. Perhaps this aspect should be reflected at least in the study's limitations, since the term primary chronic musculoskeletal pain is a broader concept and encompasses other types of conditions.

Authors' response: We thank the reviewer for this helpful comment. We agree that this aspect may limit the generalizability of the findings to other forms of chronic primary musculoskeletal pain (CPMP). Accordingly, we have revised the Limitations section to clarify that all participants in the clinical group had low back pain. The change can be found on page 9, lines 327–331.

Original text: Additionally, given the characteristics of the target population and the practical challenges of recruiting participants who met the ICD-11 criteria for CPMP, a non-probabilistic sampling method was employed, which may have limited the representativeness of the sample.

Modified text: Additionally, given the characteristics of the target population and the difficulty in identifying cases of chronic musculoskeletal pain that met the ICD-11 criteria, which require the pain not be attributable to a known medical condition, a non-probabilistic sampling method was employed. This may have limited the representativeness of the sample, as all participants in the clinical group had low back pain.